# AMPK deficiency in smooth muscles causes persistent pulmonary hypertension of the new-born and premature death

Javier Moral-Sanz ®[1], Sophronia A. Lewis[1], Sandy MacMillan ®[1], Marco Meloni[2], Heather McClafferty[1], Benoit Viollet ®[3], Marc Foretz ®[3], Jorge del-Pozo ®[4] & A. Mark Evans ®[1] ✉

AMPK has been reported to facilitate hypoxic pulmonary vasoconstriction but, paradoxically, its deficiency precipitates pulmonary hypertension. Here we show that AMPK-α1/α2 deficiency in smooth muscles promotes persistent pulmonary hypertension of the new-born. Accordingly, dual AMPK-α1/α2 deletion in smooth muscles causes premature death of mice after birth, associated with increased muscularisation and remodeling throughout the pulmonary arterial tree, reduced alveolar numbers and alveolar membrane thickening, but with no oedema. Spectral Doppler ultrasound indicates pulmonary hypertension and attenuated hypoxic pulmonary vasoconstriction. Age-dependent right ventricular pressure elevation, dilation and reduced cardiac output was also evident. $K_V1.5$ potassium currents of pulmonary arterial myocytes were markedly smaller under normoxia, which is known to facilitate pulmonary hypertension. Mitochondrial fragmentation and reactive oxygen species accumulation was also evident. Importantly, there was no evidence of systemic vasculopathy or hypertension in these mice. Moreover, hypoxic pulmonary vasoconstriction was attenuated by AMPK-α1 or AMPK-α2 deletion without triggering pulmonary hypertension.

We recently confirmed that the AMP-activated protein kinase (AMPK) facilitates hypoxic pulmonary vasoconstriction (HPV)[1–3], which aids ventilation-perfusion matching at the lungs by diverting blood from oxygen-deprived to oxygen-rich areas, but also contributes to hypoxic pulmonary hypertension when hypoxia is widespread during disease (e.g. chronic obstructive pulmonary disease) and ascent to altitude[4]. In stark contrast, a growing body of evidence suggests that AMPK deficiency may facilitate hypoxic and idiopathic pulmonary hypertension[5–7], where mechanisms are largely unknown and therapies poor[8–11]. It is important, therefore, to determine the nature of this paradox.

AMPK is a cellular energy sensor that comprises one of two catalytic α subunits, in combination with one each of the two β and three γ regulatory subunits, which together provide at least 12 heterotrimeric subunit combinations[12]. Binding of AMP to the γ subunit increases AMPK activity 10-fold by allosteric activation, with binding of AMP or ADP delivering 100-fold further activation through increased phosphorylation by liver kinase B1 (LKB1) of Thr172 on the α subunit, allied with decreased dephosphorylation by phosphatases. Each of these effects are antagonised by ATP[12]. There are also three AMP- and ADP-independent pathways to AMPK activation: long chain fatty acid regulation through the allosteric drug and metabolite (ADaM) site[13]; glucose modulation through a fructose-1,6-bisphosphate (FBP) sensing mechanism that may involve the aldolase-v-ATPase-Ragulator complex on lysosomes[14–16]; calcium-dependent activation through calcium-

[1]Centre for Discovery Brain Sciences and Cardiovascular Science, College of Medicine and Veterinary Medicine, Hugh Robson Building, University of Edinburgh, Edinburgh EH8 9XD, UK. [2]Centre for Cardiovascular Science, Queen's Medical Research Institute, University of Edinburgh, Edinburgh EH16 4TJ, UK. [3]Université Paris Cité, CNRS, INSERM, Institut Cochin, F-75014 Paris, France. [4]R(D)SVS, University of Edinburgh Easter Bush Campus, EH25 9RG, Roslin, Edinburgh, UK. ✉e-mail: mark.evans@ed.ac.uk

calmodulin activated kinase kinase 2 (CaMKK2)[17]. Once activated the classical action of AMPK is to maintain cell-autonomous metabolic homeostasis by inhibiting anabolic and activating catabolic pathways. AMPK achieves this goal, in part, by contributing to the governance of autophagy[15], mitochondrial biogenesis and integrity[18,19], and the balance between mitochondrial oxidative phosphorylation and glycolysis[20].

Recent evidence strongly supports the view that HPV is primarily facilitated through activation of AMPK-α1 containing isoforms by the canonical energy stress pathway, consequent to inhibition of mitochondrial oxidative phosphorylation[21,22] and increases in cellular AMP:ATP and ADP:ATP ratios[1–3]. Briefly, smooth muscle-selective deletion of the gene encoding AMPK-α1 (*Prkaa1*) and ~90% global hypomorphic expression of the gene that encodes LKB1 (*Stk11*) virtually abolished hypoxia-evoked increases in pulmonary vascular resistance in the mouse lung in-situ, which remained unaffected after CaMKK2 deletion[2]. By contrast, data suggest that AMPK-α2 deficiency in endothelia is associated with pulmonary hypertension of the newborn[11,23–25] and adult[5,26,27], which may facilitate the progression of pulmonary hypertension, at least in part, by inhibiting angiotensin converting enzyme 2. Accordingly, pulmonary hypertension of the newborn and adult is ameliorated by metformin[5,23,25–28], which can activate AMPK[29] indirectly by reducing the efficiency of redox and electron transfer by mitochondrial complex I[30], but may also inhibit fructose-1,6-bisphosphatase independent of AMPK[31].

One plausible explanation for this paradox could be that smooth muscle AMPK-α1 and endothelial AMPK-α2 exert distinct actions, the former promoting HPV and acute hypoxic pulmonary hypertension while the latter opposes the development of chronic pulmonary hypertension. Alternatively, HPV and persistent pulmonary hypertension may be mediated through distinct mechanisms.

We show here that dual AMPK-α1/α2 deletion in smooth muscles precipitates pulmonary hypertension and right ventricular dilation after birth, leading to premature death by 12 weeks of age. The pathological markers presented are most consistent with the characteristics of persistent pulmonary hypertension of the new-born (PPHN) rather than either hypoxic pulmonary hypertension or idiopathic persistent pulmonary hypertension of the adult.

## Results

Because global, dual AMPK-α1/α2 deletion is embryonic lethal we developed mice in which AMPK-α1 and/or AMPK-α2 catalytic subunits were selectively deleted in smooth muscles (Fig. 1a, b and Supplementary Fig. 1) as described previously[2], by crossing mice in which the sequence encoding the catalytic site of the AMPK-α1 and/or AMPK-α2 subunits was flanked by loxP sequences[32–34] with mice expressing Cre recombinase conditional on the transgelin promoter[35]. Importantly, transgelin-Cre mice do not exhibit Cre expression in endothelial cells[35]. Transgelin does, however, exhibit transient developmental expression in atrial and ventricular myocytes[35], although no loss of AMPK-α1 or AMPK-α2 protein was revealed for the atria or ventricles by Western blot (Supplementary Fig. 2).

### Dual AMPK-α1/α2 deletion in smooth muscles leads to premature death

Unexpectedly, all mice with dual, homozygous deletion of AMPK-α1/α2 died between 7 and 12 weeks after birth (68 ± 5 days, mean ± SEM; range = 33–98 days). Therefore, all AMPK-α1/α2 knockouts had to be obtained by heterozygous crosses (e.g., *Prkaa1*^flx/flx^ + *Prkaa2*^flx/flx^ + Cre^+/−^). This yielded 1 viable AMPK-α1/α2 knockout in every 50 live births (birth rate ≈2%, suggesting frequent foetus resorption by Dams). Over 5 years we obtained a total of 47 AMPK-α1/α2 knockouts, 19 males and 28 females. Of these 14 mice died prematurely before experiments could be completed. Figure 1c shows a Kaplan–Meyer curve for a prospective study comparing survival times of four mice from the same litter, while Fig. 1d

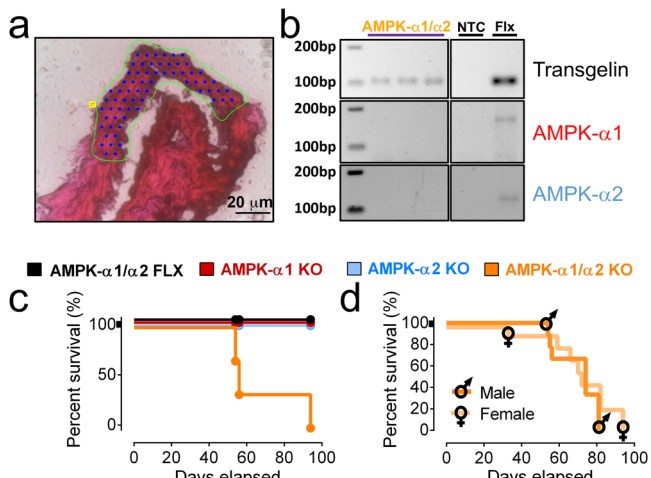

**Fig. 1 | AMPK-α1/α2 deletion precipitates premature death. a** Exemplar image of a section of pulmonary arterial media excised by laser microdissection. **b** Exemplar gels for quantitative RT-PCR amplicons for transgelin, AMPK-α1 and AMPK-α2 taken from sections of the medial layer excised from pulmonary arterial sections from AMPK-α1/α2 double knockout (KO) and AMPK-α1/α2 floxed (FLX) mice by laser microdissection. NTC negative template control (eluate from the RNA extraction for which no sample material was added). **c** Kaplan–Meyer curves for AMPK-α1/α2 FLX (black *n* = 4, 2 males and 2 females), AMPK-α1 knockouts (AMPK-α1 KO; red, *n* = 6, 3 males and 3 females), AMPK-α2 knockouts (AMPK-α2 KO; blue, *n* = 7, 2 males and 5 females) and AMPK-α1/α2 KO (orange, *n* = 4, 2 males and 2 females). **d** Retrospective analysis comparing survival rates for AMPK-α1/α2 KO males (*n* = 6) and females (*n* = 10).

shows a Kaplan–Meyer curve for retrospective analysis of all unexpected deaths of AMPK-α1/α2 knockouts by gender. Notably, there was no significant difference in survival rates between males (64 ± 6 days) and females (75 ± 7 days). Moreover, there was no evidence of sexual dimorphism with respect to any aspect of the pathologies reported below. By contrast, mice with conditional deletion of either AMPK-α1 or AMPK-α2 alone were long-lived and exhibited no obvious pathology. We therefore assessed cohorts of AMPK-α1/α2 knockout mice by necropsy (autopsy) after "natural death" and at defined time points when mice were sacrificed for experiments. Histological outcomes were compared (double blind) with age-matched controls (AMPK-α1/α2 floxed mice), AMPK-α1 and AMPK-α2 knockouts.

### Dual AMPK-α1/α2 deletion precipitates cardiopulmonary remodelling

Dual deletion of AMPK-α1/α2 had a profound impact on the cardiopulmonary system after birth. In non-terminal (Supplementary Fig. 3) and terminal samples (Fig. 2a, b; for higher resolution images see Supplementary Fig. 3) we found extensive evidence of cardiomyopathy. Diffuse thinning of left, but not right, ventricular walls of AMPK-α1/α2 knockouts was apparent (Fig. 2a, b). This was associated with multifocal cardiomyocyte thinning, with concurrent mild interstitial fibrosis of the subendocardial myocardium evident in the left, but not right, ventricular myocardium (Fig. 2aii; for higher resolution images see Supplementary Fig. 3). Greater levels of Picrosirius Red (PRS; Supplementary Fig. 4) staining were identified in AMPK-α1/α2 knockouts when compared to controls in all areas examined (Fig. 2biv; Left Ventricle, Right Ventricle, Septum; Kruskal–Wallis, *P* = 0.0133), but no increase in left ventricular weight to body weight ratio was evident (Fig. 2biii). By contrast, AMPK-α1/α2 knockouts exhibited marked increases in right ventricular weight to body weight ratio (1.05 ± 0.073) when compared to controls (0.83 ± 0.028; *P* = 0.0181; Fig. 2bii), that translated into increases in the Fulton index (RV/[LV + S]) from 0.227 ± 0.005 for controls (*n* = 10) to 0.305 ± 0.018 (*n* = 13, *P* < 0.0001;

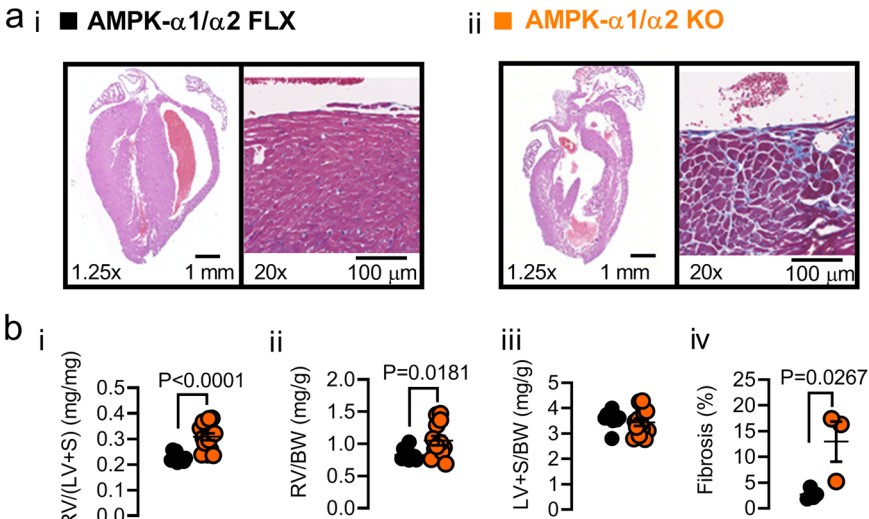

**Fig. 2 | AMPK-α1/α2 deletion precipitated remodeling of the heart.**
**a** Representative sub-gross images of heart slices from terminal samples stained with Hematoxylin–Eosin (left) and a higher magnification image showing the right aspect of the septal myocardium including right ventricular endocardium stained with Masson's trichrome (right) for (i) AMPK-α1/α2 floxed (AMPK-α1/α2 FLX) and (ii) AMPK-α1/α2 knockouts (AMPK-α1/α2 KO). **b** Scatter plots show the (i) Fulton index [RV/(LV + S) ratio], (ii) right ventricular weight relative to body weight ratio (RV/BW), and (iii) left ventricular plus septum weight to body weight ratio (LV + S/BW) and (iv) fibrosis of the right ventricle for AMPK-α1/α2 floxed (n = 10 fields of view per mouse from n = 4 mice) and AMPK-α1/α2 KO (n = 13 fields of view per mouse from n = 3 mice). Data are expressed as mean ± SEM. Statistical significance was assessed by two-sided unpaired Student's t test.

Fig. 2bi). Despite these structural changes, AMPK-α1/α2 double knockouts did not exhibit pathological alterations in any of the valves, microvasculature, endocardium, epicardium, or major vessels examined. In short, right ventricle dilation and increases right ventricular weight to body weight ratio occurred in the absence of right-sided ventricular thinning, indicating cardiac eccentric hypertrophy that persists and could therefore be driven, at least in part, by pulmonary hypertension[11,36,37].

Accordingly, in terminal samples of distal pulmonary arteries we measured marked thickening of the medial layer (Fig. 3ai-ii; see also Supplementary Fig. 5) that was coupled with pronounced increases in muscularisation (Fig. 3aiii and Supplementary Fig. 5 for measures by total vessel counts; for higher resolution images see Supplementary Fig. 6 (terminal) and Fig. 6c (non-terminal)). This was exemplified by the medial thickness of small pulmonary arteries (≈45 μm external diameter) stained with α-smooth muscle actin, which measured 4.25 ± 0.28 μm²/μm in AMPK-α1/α2 knockouts (122 vessels, n = 4 mice) compared to 1.59 ± 0.16 μm²/μm for controls (AMPK-α1/α2 floxed; 141 vessels, n = 4 mice, P < 0.0001). By contrast no significant difference in medial thickness was noted for either age-matched AMPK-α1 (1.29 ± 0.20 μm²/μm, 122 vessels, n = 4 mice) or AMPK-α2 knockouts (1.36 ± 0.25 μm²/μm, 136 vessels, n = 4 mice). Importantly, there was also no difference between groups with respect to either external diameter or the number of resistance-sized arteries per field (Fig. 3bi-ii). Nor was there any evidence of intimal fibrosis, hyperplasia or plexiform lesions. Therefore, the pulmonary vasculopathy observed here is primarily associated with increased medial thickness throughout the pulmonary arterial tree, with muscle evident within pulmonary arteries that would not normally be muscular. It was therefore surprising that we identified little evidence of mitotic activity in the medial layer of pulmonary arteries from AMPK-α1/α2 knockouts, levels of Ki67 labelling being comparable to controls (Supplementary Fig. 7), and there was no evidence of cellular shrinkage or nuclear fragmentation that would be expected if there was extensive apoptosis[38,39]. This suggests that increased muscularisation was consequent to smooth muscle cell hypertrophy rather than proliferation. Importantly, there was no evidence of parenchymal fibrosis (Fig. 3ci-ii and Supplementary Fig. 8) or lung oedema (Fig. 3d), indicating that pulmonary vasculopathy arose independent of either left ventricular dysfunction or

chronic congestive heart failure[40]; note, analysis of PRS staining indicated that parenchymal fibrosis was slightly higher in terminal samples of AMPK-α1/α2 knockouts, but this did not reach significance (Supplementary Fig. 8). By contrast and despite AMPK-α1/α2 deletion occurring in all smooth muscles, histological assessment of the systemic vasculature in terminal samples with haematoxylin/eosin and α-smooth muscle actin revealed no evidence of hepatic centrilobular congestion or fibrosis, arteriolar hyalinosis or hypertrophy within the tunica media or intima of renal arteries from AMPK-α1/α2 knockouts (Fig. 3ei-ii), nor vasculopathy or any other evident lesion of the brain (Supplementary Fig. 9 and Supplementary Table 2).

However, structural changes in the lungs of AMPK-α1/α2 knockouts were evident. There was simplification of alveolar spaces with reductions in alveolar numbers (Fig. 4a, b; AMPK-α1/α2 floxed 290 ± 24 alveoli/mm² vs non-terminal AMPK-α1/α2 knockouts 201 ± 23 (P = 0.0317) and terminal AMPK-α1/α2 knockouts 143 ± 19 alveoli/mm² (P = 0.0286)), that was accompanied by thickening of the alveolar wall (Fig. 3c, d; AMPK-α1/α2 floxed 5.7 ± 0.1 μm vs non-terminal AMPK-α1/α2 knockouts 7.4 ± 0.1 μm (P = 0.0286) and terminal AMPK-α1/α2 knockouts 9.5 ± 0.2 μm (P = 0.0286).

We therefore assessed lung sections from neonates (P10) prior to alveolarisation[41–43]. As one would expect, when compared to alveoli in lung sections from adults (Fig. 4) the alveoli of P10 AMPK-α1/α2 floxed mice were fewer in number and had thicker walls (P = 0.0159; two-sided unpaired Mann–Whitney's test). Importantly, there was no significant difference in either alveolar number or alveolar wall thickness between neonatal (P10) AMPK-α1/α2 floxed mice and neonatal (P10) AMPK-α1/α2 knockouts (Fig. 5a–d). However, increases in the medial thickness of pulmonary arteries were evident in AMPK-α1/α2 knockouts when compared to AMPK-α1/α2 floxed mice (n = 4 mice, P = 0.0079; Fig. 5e).

When taken together, therefore, histological data on mice with AMPK-α1/α2 deletion targeted to smooth muscles are most consistent with persistent pulmonary hypertension of the new-born (PPHN), which is associated with alveolar simplification, reduced alveolar number and alveolar wall thickening, together with greater pulmonary vascular hypertrophy, less intimal fibrosis and fewer plexiform lesions[41–43] than observed in adults with pulmonary hypertension[38]; although recent observations point to a transient hyperproliferative

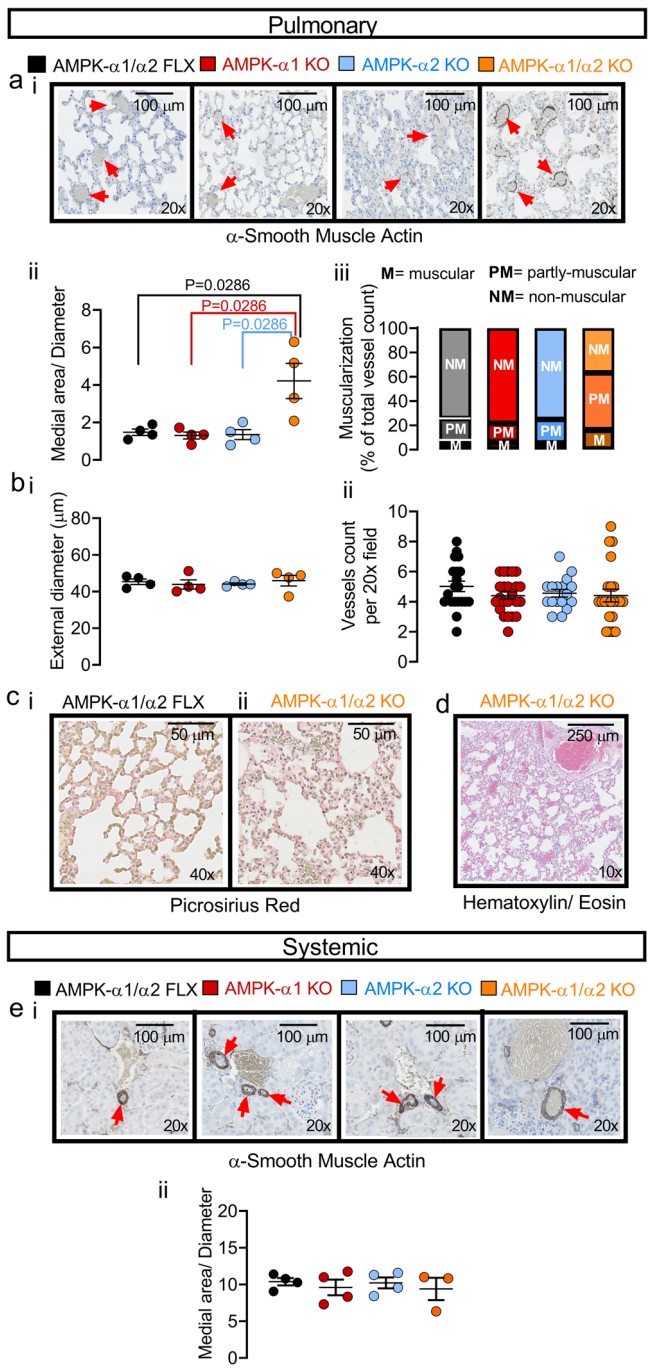

**Fig. 3 | AMPK-α1/α2 deletion precipitated remodeling of pulmonary arterial but not systemic arterial system. a** Representative images of lung slices from terminal samples stained for (i) α-smooth muscle actin, together with scatter plots of (ii) medial area corrected by diameter and (iii) degree of muscularisation for all pulmonary arteries analysed for AMPK-α1/α2 knockouts (AMPK-α1/α2 KO; n = 4 mice, average of 22–39 arteries per mouse, 7 fields per mouse) after death at 7–10 weeks, and age-matched AMPK-α1/α2 floxed (AMPK-α1/α2 FLX, n = 4 mice, average of 30–39 arteries per mouse, 7 fields per mouse), AMPK-α1 KO (n = 4 mice, average of 25–37 arteries per mouse, 7 fields per mouse) and AMPK-α2 KO (n = 4 mice, average of 30–39 arteries per mouse, 7 fields per mouse). **b** Scatter plots show the mean ± SEM for the (i) external diameter (n = 4 mice per genotype) and (ii) number of vessels found per 20x field for the analysis shown in (a) (n = 28 fields from n = 4 mice per genotype, 7 fields per mouse). Images of (c) picrosirius red staining and (d) Hematoxylin–Eosin staining of lung slices show the absence of parenchymal/alveolar remodeling and the lack of lung oedema, respectively. **e** (i) representative images of renal slices stained for α-smooth muscle actin, together with (ii) plots of the medial area corrected by diameter analysed for AMPK-α1/α2 KOs (n = 4 mice, average of 20–34 arteries per mouse, 4 fields per mouse) and age-matched AMPK-α1/α2 floxed (n = 4 mice, average of 17–30 arteries per mouse, 4 fields per mouse), AMPK-α1 KOs (n = 4 mice, average of 16–30 arteries per mouse, 4 fields per mouse) and AMPK-α2 KOs (n = 3 mice, average of 25–28 arteries per mouse, 7 fields per mouse). For clarity, only mean values are presented in panel aiii. The rest of the panels are expressed as mean ± SEM. Statistical significance was assessed by two-sided unpaired Mann–Whitney's test (aii, bi-ii, eii) or a Friedman test with Dunn's correction for multiple comparisons (aiii).

normal range of 786 ± 26 mm/s and 2.7 ± 0.3 cm for AMPK-α1/α2 floxed (n = 13; P < 0.0001; Fig. 6ai–iii and Supplementary Fig. 10a). Strikingly, induction of hypoxic pulmonary vasoconstriction (HPV) and thus acute hypoxic pulmonary hypertension by mild hypoxia (8% $O_2$) was virtually abolished in all AMPK-α1/α2 knockouts, SPV declining by only −77 ± 7 mm/s (n = 4) during 2 min exposures to 8% $O_2$, compared to a net fall of −246 ± 27 mm/s for controls (n = 7, P = 0.006; Fig. 6aiv and Supplementary Fig. 10b). This is entirely consistent with previous studies that showed that AMPK-α1 deletion blocked HPV, while AMPK activation mimicked HPV[1,46]. Moreover, it is consistent with reports of reduced HPV following development of chronic hypoxic pulmonary hypertension[47,48].

The possibility that AMPK-α1/α2 deletion precipitated pulmonary hypertension after birth gained further support from direct measurement of right ventricular pressures at 7–10 weeks of age (Fig. 6b). For control mice (AMPK-α1/α2 floxed) right ventricular diastolic pressure (RVDP) measured 0.7 ± 0.5 mmHg and right ventricular systolic pressure (RVSP) 6.7 ± 0.9 mmHg (n = 9). These values were consistent across all control mice (Fig. 6biii), confirming reproducibility under our experimental conditions. It is notable, however, that values measured are at the lower end of the range of RVSP available in the literature[49,50]. This could be due to the experimental approaches used here but may also reflect the fact pulmonary vascular resistance falls markedly after birth in humans, pigs and rodents[42,43,50], and takes months to recover to adult levels in humans. Nevertheless, it was evident that AMPK-α1/α2 deletion led to age-dependent increases in RVDP and RVSP after birth (Fig. 6b and Supplementary Fig. 11). Despite the small number of animals available (n = 5), on average these values for AMPK-α1/α2 knockouts translated into significant increases in RVSP (7–10 weeks 17.7 ± 4.1 mmHg (P = 0.056); 8–10 weeks, 21.3 ± 2.3 mmHg (P = 0.0028)) but not RVDP (7–10 weeks, 9.5 ± 4.4 mmHg (NS, P = 0.2398); 8–10 weeks, 11.9 ± 4.7 mmHg (NS, P = 0.1483)) relative to controls (Fig. 6bii-iv). Accordingly, and consistent with findings for terminal samples, histological analysis of lungs from these mice used for experimentation revealed that the main intrapulmonary artery and branches of all AMPK-α1/α2 knockouts exhibited marked remodelling, the primary features of which were thickened tunica media and adventitia, and disorganised hypercellularity in both tunica media and adventitia (Fig. 6ci-ii).

state early in the course of the disease in adults too, with smooth muscle senescence and resistance to apoptosis predominant at more advanced stages[38,44].

## Dual AMPK-α1/α2 deletion leads to pulmonary hypertension after birth

We assessed pulmonary vascular function in anesthetised AMPK-α1/α2 knockouts using spectral Doppler ultrasound analysis of systolic peak velocity (SPV) and velocity time integral (VTI) within the main pulmonary artery (this method provides an index of dynamic changes in pulmonary vascular resistance, while avoiding confounding variables due to enhanced cardiac output[45]). During normoxia SPV and VTI were markedly lower for AMPK-α1/α2 knockouts (7–10 weeks) than for age-matched controls (AMPK-α1/α2 floxed mice), indicating higher pulmonary vascular resistance, measuring, respectively, 552 ± 17 mm/s and 1.7 ± 0.1 cm for AMPK-α1/α2 knockouts (n = 10) compared to the

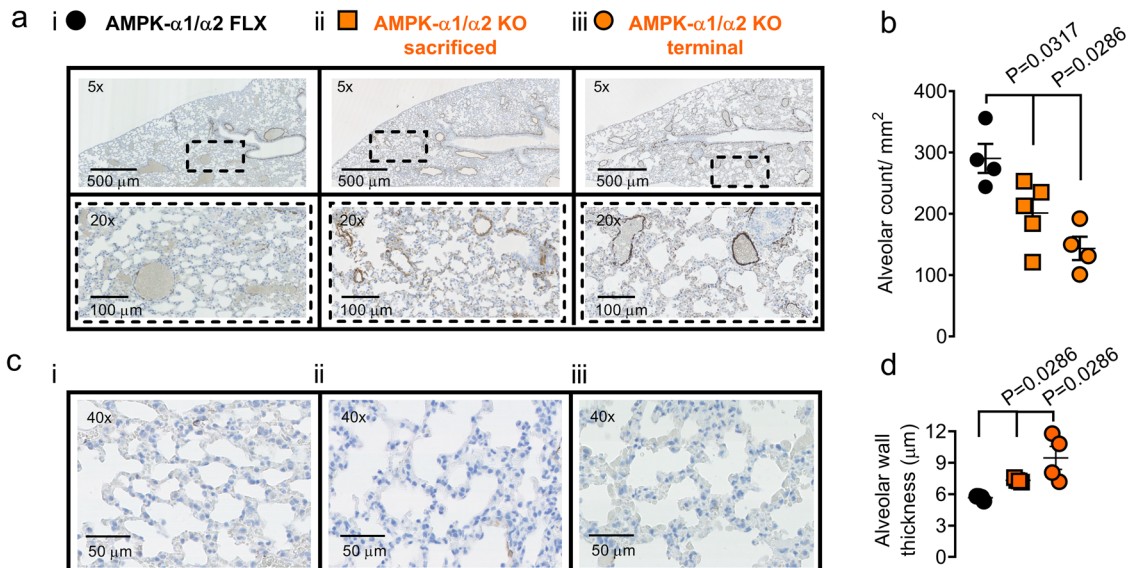

**Fig. 4 | AMPK-α1/α2 deletion reduces alveolar number and precipitates thickening of alveolar walls. a** Representative images of alveoli in lung slices from terminal samples stained for α-smooth muscle actin. **b** Scatter plot shows alveolar number per mm² for terminal ($n = 4$ mice) and non-terminal ($n = 5$ mice) samples of AMPK-α1/α2 knockouts (AMPK-α1/α2 KO) vs age-matched AMPK-α1/α2 floxed controls (AMPK-α1/α2 FLX; $n = 4$ mice). The analysis in (**b**) was averaged from 2 mm² area per mouse. **c** Representative images of alveoli in lung slices from terminal samples stained for α-smooth muscle actin. **d** Scatter plot shows alveolar wall thickness for terminal ($n = 4$) and non-terminal ($n = 4$) samples of AMPK-α1/α2 KO vs age-matched AMPK-α1/α2 FLX ($n = 4$). The analysis in d was averaged from 60–70 septa per mouse. Data are expressed as mean ± SEM. Statistical significance was assessed by two-sided unpaired Mann–Whitney's test.

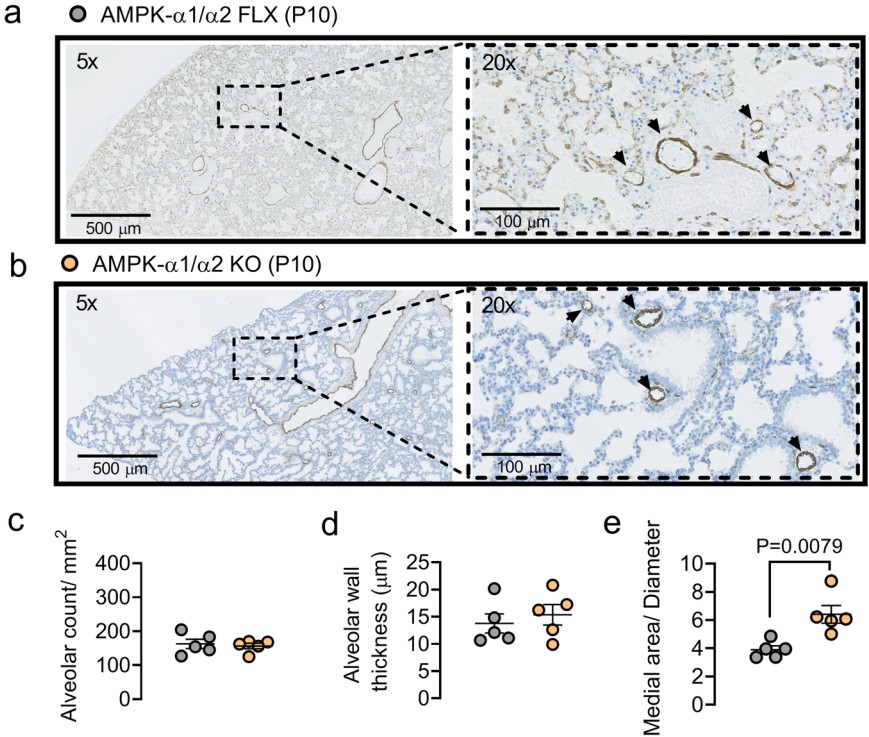

**Fig. 5 | In neonatal lungs AMPK-α1/α2 deletion has no effect on alveolar number or wall thickness but increases medial thickness of pulmonary arteries.** Representative images of alveoli in lung slices from non-terminal samples stained for α-smooth muscle actin from (**a**) age-matched AMPK-α1/α2 floxed (AMPK-α1/α2 FLX) controls and (**b**) AMPK-α1/α2 KOs. Scatter plots show (**c**) alveolar number per mm² ($n = 5$ mice per genotype, averaged from 2 mm² area per mouse), (**d**) alveolar wall thickness ($n = 5$ mice per genotype, averaged from 70–90 septa per mouse) and (**e**) medial thickness of pulmonary arteries for non-terminal samples of AMPK-α1/α2 KO (orange, $n = 5$ mice, $n = 4$ fields, averaged from 16–25 arteries per mouse) vs age-matched AMPK-α1/α2 FLX (gray, $n = 5$ mice, $n = 4$ fields, averaged from 16–22 arteries per mouse). Data are expressed as mean ± SEM. Statistical significance was assessed by two-sided unpaired Mann–Whitney's test.

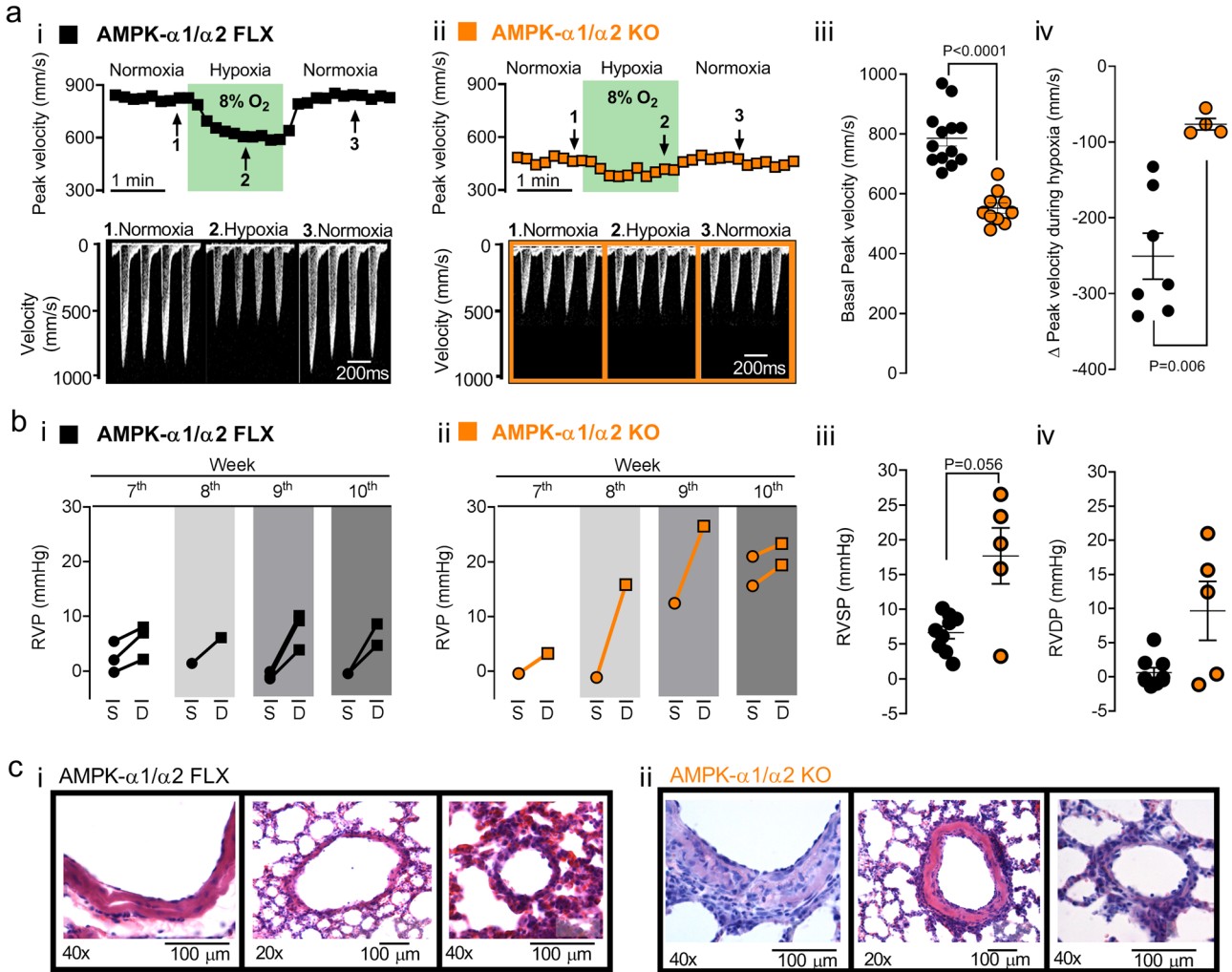

**Fig. 6 | AMPK-α1/α2 deletion increased pulmonary vascular resistance and attenuated hypoxic pulmonary vasoconstriction after birth. a** i-ii Upper panels show representative spectral Doppler peak systolic velocities versus time within the main pulmonary artery of (i) AMPK-α1/α2 floxed (FLX) and (ii) AMPK-α1/α2 KO mice during normoxia, hypoxia (8% O$_2$) and recovery; lower panels show example records of Doppler velocity during normoxia, hypoxia (shaded green) and recovery. Scatter plots show the mean ± SEM for the (iii) basal peak velocity under normoxia for AMPK-α1/α2 FLX ($n = 13$ mice) and AMPK-α1/α2 KO ($n = 10$ mice) and (iv) the maximum change in peak velocity observed during 8% O$_2$ for AMPK-α1/α2 FLX ($n = 7$ mice) and AMPK-α1/α2 KO mice ($n = 4$ mice). **b** Graphs illustrate age-

dependent changes in right ventricular systolic (S) and diastolic (D) pressures for (i) AMPK-α1/α2 floxed and (ii) AMPK-α1/α2 KO mice (different weeks identified by different shades of gray). Scatter plots show values of right ventricular (iii) systolic and (iv) diastolic pressures for AMPK-α1/α2 FLX ($n = 9$) and AMPK-α1/α2 KO ($n = 5$). **c** Representative images of lung slices stained with Hematoxylin−Eosin, showing an intralobar artery (left), medium sized pulmonary artery (middle) and arteriole (right) for (i) AMPK-α1/α2 FLX ($n = 9$) and (ii) AMPK-α1/α2 KO euthanised at -80 days ($n = 5$). Data are expressed as mean ± SEM and statistical significance was assessed by two- sided unpaired Student's $t$ test (aiii) and two- sided unpaired Mann−Whitney's test (aiv, biii-iv).

Spectral Doppler ultrasound also revealed that AMPK-α1/α2 deletion triggered reductions in both right and left ventricular fractional shortening after birth, from 45 ± 4 % and 41 ± 2% for controls ($n = 4$) to, respectively, 18 ± 5 % and 9 ± 2% for AMPK-α1/α2 knockouts ($n = 5$, Fig. 7ai-iv and Supplementary Table 1). Dilation of the right atrial appendage and the left ventricle was also observed (Fig. 7b, c and Supplementary Table 1). Reduced right ventricular fractional shortening is most likely explained by increases in pulmonary vascular resistance, because there was no evidence of right ventricular thinning. Accordingly, we observed a significant age-dependent decrease in cardiac output in AMPK-α1/α2 knockouts when compared to AMPK-α1/α2 floxed mice (Supplementary Fig. 12). By contrast, reductions in left ventricular fractional shortening were not consequent to systemic arterial hypertension (Fig. 8) but could be due to reductions in pre-load consequent to reduced pulmonary venous return and/or diffuse thinning of the left ventricular walls and interstitial fibrosis identified

above (Fig. 2aii and biv, and Supplementary Fig. 3). In this respect it is important to note that left, but not right, ventricular dysfunction was also observed for AMPK-α2 knockouts, where reduced cardiac output (21.1 ± 0.3 ml min$^{-1}$, $n = 4$, $P = 0.0286$) and stroke volume (40.2 ± 1.1 μl), $n = 4$, $P = 0.0286$) were noted relative to controls (cardiac output, 28.5 ± 0.9 ml/min; stroke volume, 51.9 ± 1.8 μl; Supplementary Table 1), in agreement with previous studies that identified a predominant protective role of AMPK-α2 over AMPK-α1 in the heart[51]. When taken together these data provide indirect support for the view that right ventricular myopathy of AMPK-α1/α2 knockouts is likely driven, at least in part, by the onset of pulmonary hypertension after birth (see Discussion for further details), a view that gains further indirect support from the fact that we found no evidence of differences in either blood vessel number ($n = 5$; Note, two-sided unpaired Mann−Whitney $P = 0.056$) or medial thickness ($n = 5$) for the right ventricle of AMPK-α1/α2 knockouts when compared to controls (Supplementary Fig. 13).

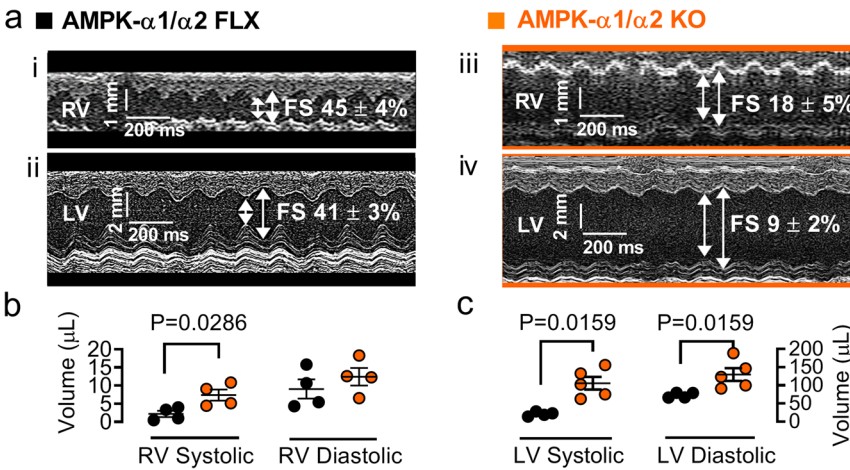

**Fig. 7 | AMPK-α1/α2 deletion precipitated reduced right and left ventricular shortening and volumes. a** i-iv Spectral Doppler parasternal long axes obtained using M-mode analysis of the (i and iii) right ventricle (RV), (ii and iv) left ventricle (LV). Dot plots show associated measures of (**b**) RV and (**c**) LV volumes for AMPK-

α1/α2 floxed controls (AMPK-α1/α2 FLX, $n = 4$ mice) and AMPK-α1/α2 knockout (AMPK-α1/α2 KO, $n = 4$ mice for RV volumes and $n = 5$ mice for LV volumes). Data are expressed as mean ± SEM. Statistical significance was assessed by two-sided unpaired Mann–Whitney's test.

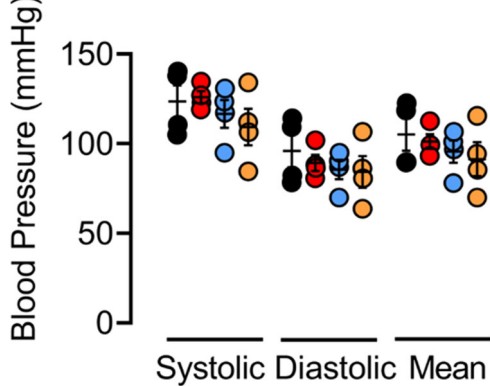

**Fig. 8 | AMPK-α1/α2 deletion in smooth muscles had little or no effect on systemic arterial blood pressures.** Dot plots show mean ± SEM for systolic, diastolic and mean systemic arterial blood pressures under resting conditions for AMPK-α1/α2 floxed (AMPK-α1/α2 FLX, black, $n = 4$ mice), AMPK-α1 knockout (AMPK-α1 KO, red, $n = 4$ mice), AMPK-α2 knockout (AMPK-α2 KO, blue, $n = 4$ mice) and AMPK-α1/α2 knockout (AMPK-α1/α2 KO, orange, $n = 4$ mice). Statistical significance was assessed by Kruskall Wallis' test with Dunn's correction for multiple comparisons.

## AMPK-α1/α2 deletion reduces $K_V1.5$ in pulmonary arterial myocytes

We next investigated the impact of AMPK-α1/α2 deletion on voltage-gated potassium ($K_V$) currents in acutely isolated pulmonary arterial smooth muscle cells, because reduced activity and/or expression of $K_V1.5$ has been identified as a hallmark of persistent pulmonary hypertension in neonates[52,53] and pulmonary hypertension in adults[54,55], where it has been proposed to decrease $K^+$ efflux and thus oppose apoptosis[55–58]. Under normoxia $K_V$ current density was markedly reduced in acutely isolated pulmonary arterial myocytes from AMPK-α1/α2 knockouts (Fig. 9a, b), measuring $5.6 ± 0.6$ pA/pF ($n = 12$) compared to $11.6 ± 0.9$ pA/pF for controls at 0 mV (AMPK-α1/α2 floxed, $n = 11$, $P < 0.0001$). Furthermore, we observed no reduction in $K_V$ current amplitude during hypoxia in myocytes from AMPK-α1/α2 knockouts (Supplementary Fig. 14), which was in-line with expectations given our previous finding that AMPK-α1 deletion blocked hypoxia-evoked reductions in $K_V1.5$ currents and HPV[2]. That reductions in normoxic $K_V$ current magnitude by AMPK-α1/α2 deletion were due to loss of $K_V1.5$ in particular was demonstrated by reduced sensitivity of available $K_V$

currents to inhibition by the specific $K_V1.5$ channel blocker DPO-1 (1 μmol/L; Fig. 9a-c). By contrast, $K_V1.5$ current magnitude in pulmonary arterial myocytes from AMPK-α1 and AMPK-α2 knockouts was found to be equivalent to controls during normoxia[2]. The discovery of reduced normoxic $K_V1.5$ currents in myocytes following AMPK-α1/α2 deletion was in of itself an unexpected outcome, given that AMPK-α1 directly phosphorylates and inhibits $K_V1.5$ in these cells[2]. This was more surprising still, because we found no reduction in transcription of the gene encoding $K_V1.5$ (*KCNA5*) by qRT-PCR, expression measuring $0.94 ± 0.05$ for AMPK-α1/α2 knockouts ($n = 4$) relative to control (AMPK-α1/α2 floxed; $n = 4$). Hence, we investigated the possibility that $K_V1.5$ availability might be determined by reduced phosphorylation by AMPK of ser559 and ser592 on the $K_V1.5$ alpha subunit, by incorporating dephospho-mimetic mutations. Transient transfection of HEK293 cells with the mutant $K_V1.5^{ser559A/ser592A}$ conferred $K_V$ currents that were significantly reduced relative to $K_V1.5$ current amplitude recorded in paired HEK293 cell cultures transfected with wild type $K_V1.5$ (Fig. 9d). This presents us with a paradox given that we have previously established that AMPK directly phosphorylates ser559 and ser592 on the $K_V1.5$ α subunit and thus inhibits voltage-dependent activation of $K_V1.5$ currents during hypoxia[2]. AMPK may therefore regulate $K_V1.5$ availability by multiple, context-specific mechanisms. Indirect support for this is provided by our previous studies which showed that the dephosphomimetic mutation S559A alone confers optimal blockade of $K_V1.5$ current inhibition by AMPK, while the dephosphomimetic mutation S592A delivers more marked reductions in $K_V1.5$ phosphorylation by AMPK than S559A[2] (see Discussion for further details).

Expression of the gene encoding TASK-1 potassium channels (*Kcnk3*), which has been associated with pulmonary hypertension[59–62] and right ventricular dysfunction[63] in adults, also remained unaffected following AMPK-α1/α2 deletion, measuring $0.95 ± 0.13$ for AMPK-α1/α2 knockouts relative to controls (AMPK-α1/α2 floxed; $n = 4$). We also found that AMPK activation had no effect on potassium currents carried by TASK-1 (Supplementary Fig. 15 and Supplementary methods) and no AMPK recognition sites were identified for this channel (either by SCANSITE4, or an algorithm available on GitHub (https://github.com/BrunetLabAMPK/AMPK_motif_analyzer[64]). It is therefore unlikely that TASK-1 is subject to modulation by AMPK.

## AMPK-α1/α2 deletion triggers mitochondrial dysfunction

AMPK-α1/α2 deletion did not alter the expression of cytochrome c oxidase subunit 4i2 (COX4i2; $1.02 ± 0.15$ relative to controls (AMPK-α1/

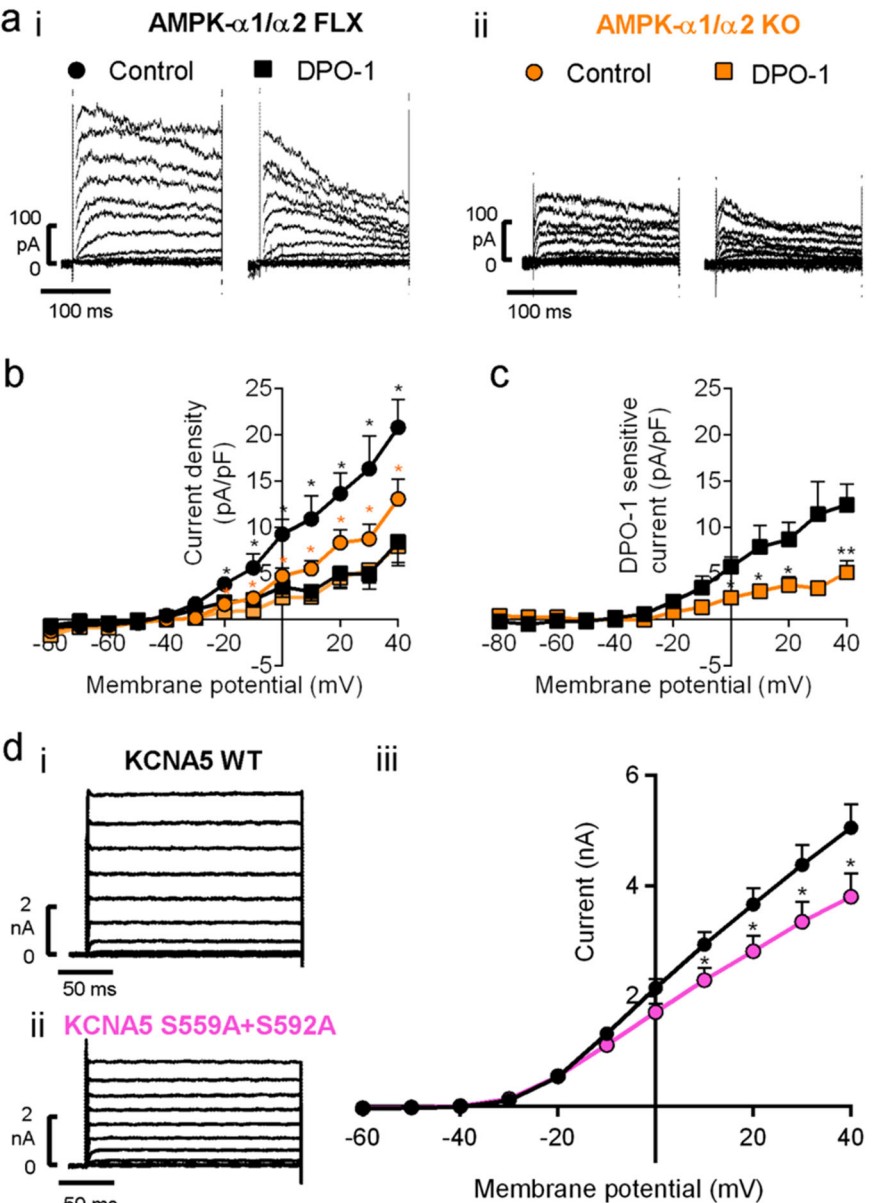

**Fig. 9 | AMPK-α1/α2 deletion reduces K_V1.5 current density in pulmonary arterial myocytes. a** Panels show example records for K_V currents recorded during control conditions and following extracellular application of 1 μM DPO-1 in acutely isolated pulmonary arterial myocytes from (i) AMPK-α1/α2 floxed (AMPK-α1/α2 FLX) and (ii) AMPK-α1/α2 knockouts (AMPK-α1/α2 KO). **b** Comparison of current-voltage relationship for K_V currents recorded in pulmonary arterial myocytes from AMPK-α1/α2 FLX (n = 4 cells, from n = 4 mice) and AMPK-α1/α2 KO mice (n = 5 cells from n = 4 mice) under control (normoxic) conditions and after extracellular application of 1μmol/L DPO-1. **c** Current-voltage relationship for DPO-1-sensitive current (current before DPO-1 minus current after DPO-1) for the experiments shown in (**b**). **d** Panels show example records for K⁺ currents in HEK293 cells transfected with (i) KCNA5 wild type (WT) or (ii) KCNA5 containing S559A + S592A dephospho-mimetic mutation at identified AMPK phosphorylation sites. **d** iii Comparison of current-voltage relationship for K_V1.5 currents recorded in HEK 293

cells transfected with KCNA5 WT (n = 16 cells from n = 5 independent transfections) or S559A + S592A dephospho-mimetic KCNA5 mutant (n = 17 cells from n = 5 independent transfections paired with WT). Data are expressed as mean ± SEM. Statistical significance was assessed by two-sided paired (**b**) and two-sided unpaired Student's *t* test (**c** and **d** iii). *P* values in (**b**) for AMPK-α1/α2 FLX vs AMPK-α1/α2 FLX + DPO-1: −20mV, * = 0.0225; −10mV, * = 0.0113; 0 mV, * = 0.0133; 10 mV, * = 0.0176; 20 mV, * = 0.0179; 30 mV, * = 0.0185; 40 mV, * = 0.0115. *P* values in (**b**) for AMPK-α1/α2 KO vs AMPK-α1/α2 KO + DPO-1: −20mV, * = 0.0291; −10mV, * = 0.0429; 0 mV, * = 0.0194; 10 mV, * = 0.0393; 20 mV, * = 0.0167; 30 mV, * = 0.0233; 40 mV, * = 0.0192. *P* values in (**c**) for AMPK-α1/α2 FLX vs AMPK-α1/α2 KO: 0 mV, * = 0.0133; 10 mV, * = 0.0117; 20 mV, * = 0.0219; 30 mV, * = 0.0850; 40 mV, ** = 0.0095. *P* values in diii for KCNA5 WT vs KCNA5. S559A + S592A: 10 mV; * = 0.0492; 20 mV, * = 0.0483; 30 mV, * = 0.0496; 40 mV, * = 0.045.

α2 floxed); *n* = 4), an atypical nuclear encoded subunit that appears critical to mitochondrial mechanisms of oxygen-sensing and thus HPV[21,22], lending further indirect support for the view that AMPK facilitates HPV and inhibits K_V1.5 downstream of mitochondria[2].

However, AMPK-α1/α2 deletion did precipitate mitochondrial dysfunction in pulmonary arterial myocytes. When compared to controls (Fig. 10ai and bi) marked depolarisation of the mitochondrial

membrane potential was evident in pulmonary arterial myocytes from AMPK-α1/α2 knockout mice (Fig. 10aii and bii), evidenced by dramatic reductions in TMRE and Rhod123 fluorescence intensities in Mito-Tracker positive mitochondria to 34.20 ± 6.19 arbitrary units (AU) and 836 ± 104 AU, respectively, relative to paired measures of 396 ± 58 AU and 1689 ± 106 AU from myocytes of AMPK-α1/α2 floxed mice (note, the more hyperpolarised the mitochondria the more of each of these

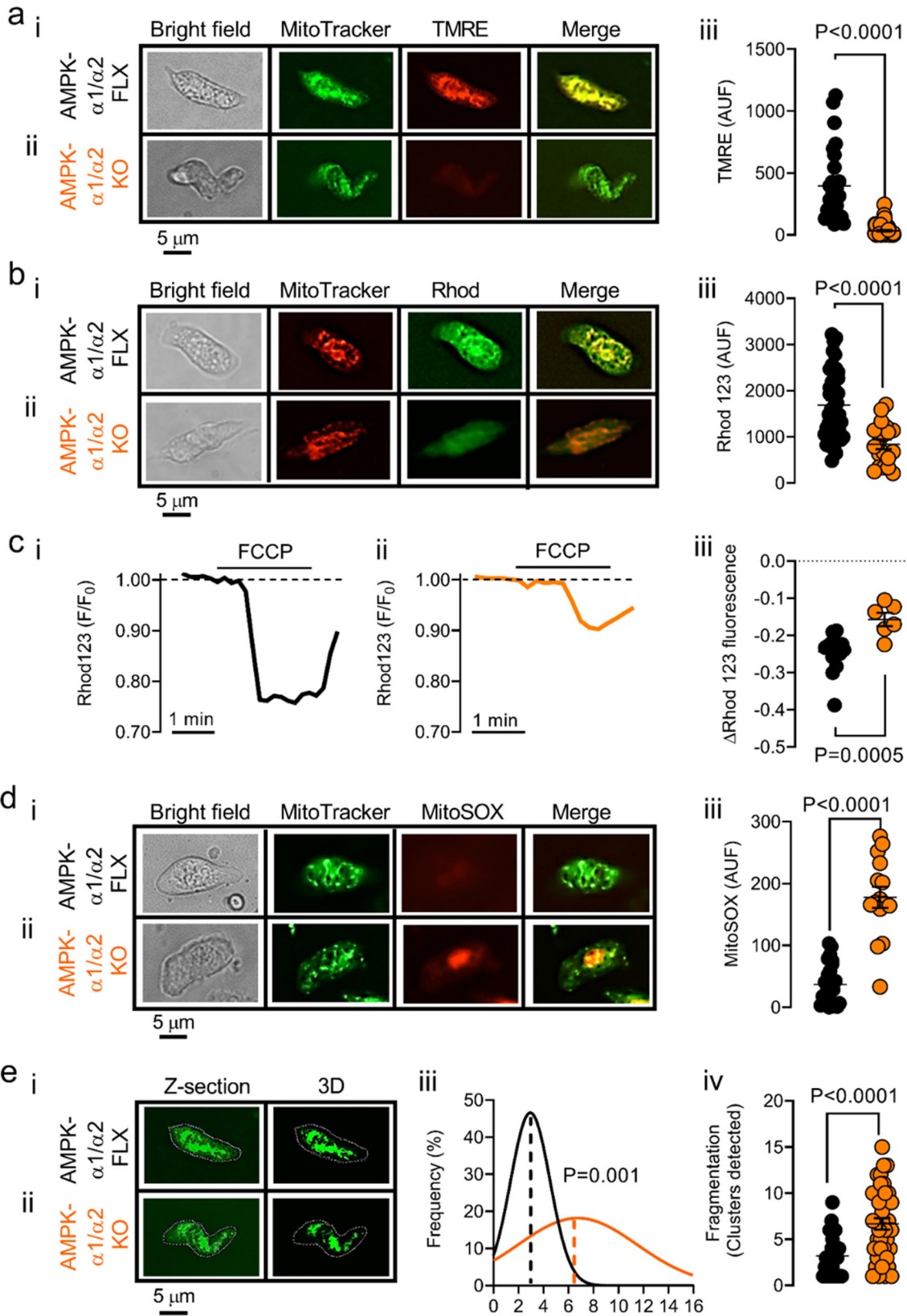

dyes is accumulated). Further support for this can be taken from the marked attenuation of mitochondrial depolarisation in myocytes from AMPK-α1/α2 knockouts in response to the mitochondrial protonophore FCCP (10 μmol/L, Fig. 10ci-iii); the response to which also served to confirm non-quenching conditions for the two dyes used here. Allied to this, co-labelling with the reactive oxygen species (ROS)

indicator MitoSOX indicated marked increases in ROS in pulmonary arterial myocytes from AMPK-α1/α2 knockout mice when compared to paired cells from controls (Fig. 10di–iii). Furthermore, analysis of the number of mitochondrial clusters per cell indicated increased fragmentation of mitochondria in pulmonary arterial myocytes from AMPK-α1/α2 knockout mice (Fig. 10ei-iv). This is in accordance with

**Fig. 10 | AMPK-α1/α2 deletion precipitates mitochondrial dysfunction and reactive oxygen species accumulation in pulmonary arterial myocytes.**
**a** Images show from left to right a bright-field image of a pulmonary arterial myocyte, then deconvolved 3D reconstructions of a z-stack of images of Mito-Tracker Green fluorescence, TMRE fluorescence and a merged image from (i) AMPK-α1/α2 floxed (AMPK-α1/α2 FLX) and (ii) AMPK-α1/α2 knockouts (AMPK-α1/α2 KO). **a** iii Scatter plot shows the mean ± SEM for total (non-deconvolved) TMRE fluorescence ($n = 26$ cells for AMPK-α1/α2 FLX from 8 independent experiments, $n = 3$ mice; $n = 54$ cells for AMPK-α1/α2 KO from 13 independent experiments, $n = 3$ mice). **b** i-ii As in A but showing images of MitoTracker Red and Rhodamine 123 fluorescence. **b** iii, Scatter plot shows the mean ± SEM for total (non-deconvolved) Rhodamine 123 fluorescence ($n = 41$ cells for AMPK-α1/α2 FLX from 10 independent experiments, $n = 5$ mice; $n = 20$ cells for AMPK-α1/α2 KO from 6 independent experiments, $n = 3$ mice). **c** Records of Rhodamine 123 fluorescence ratio ($F/F_0$) against time recorded in pulmonary arterial myocytes from (i) AMPK-α1/α2 floxed, (ii) AMPK-α1/α2 KO mice, and (iii) a scatter plot showing the mean ± SEM for the maximum change in Rhodamine 123 fluorescence during application of a mitochondrial uncoupler, FCCP (10 μmol/L; $n = 10$ cells for AMPK-α1/α2 FLX from 5 independent experiments, $n = 3$ mice; $n = 6$ cells for AMPK-α1/α2 KO from 4 independent experiments, $n = 3$ mice). **d** i-ii As in (**a**) but showing images of MitoTracker Green and MitoSOX fluorescence. **d** iii, Scatter plot shows the mean ± SEM for total (non-deconvolved) MitoSOX fluorescence ($n = 24$ cells for AMPK-α1/α2 FLX from 5 independent experiments, $n = 2$ mice; $n = 15$ cells for AMPK-α1/α2 KO from 4 independent experiments, $n = 2$ mice). **e** Images of z-sections and 3D reconstruction for the mitochondrial clusters detected in the cells shown in (**a**), from (i) AMPK-α1/α2 floxed and (ii) AMPK-α1/α2 KO, together with a (iii) frequency histogram (bin centre = 2, mean value indicated by dotted line) for the values shown in (**e** iv) and (iv) scatter plot for cluster detection ($n = 25$ cells for AMPK-α1/α2 FLX from 7 independent experiments, $n = 3$ mice; $n = 39$ for AMPK-α1/α2 KO from 9 independent experiments, $n = 4$ mice). Data are expressed as mean ± SEM. Statistical significance was assessed by two-sided unpaired Mann–Whitney's test (**a** iii, **c** iii), two-sided unpaired Student's $t$ test (**b** iii, **d** iii, **e** iv) and extra sum-of squares $F$-test for Gaussian distributions (**e** iii).

the finding that mitochondrial dysfunction and/or ROS accumulation are associated with persistent pulmonary hypertension in neonates[23,65,66], and pulmonary hypertension in adults[67–70]. Mitochondrial dysfunction in pulmonary arterial myocytes of AMPK-α1/α2 knockouts was not consequent to changes in the expression of either COX4i2 (see above) or peroxisome proliferator-activated receptor gamma coactivator 1-alpha (PGC-1α; $0.74 ± 0.13$, $n = 4$) in AMPK-α1/α2 knockouts relative to AMPK-α1/α2 floxed, but is entirely consistent with the fact that AMPK directly phosphorylates and thus regulates a plethora of targets critical to the governance of mitochondrial biogenesis, integrity and mitophagy[15,18,19].

## Discussion

The present investigation reveals a direct link between smooth muscle-selective AMPK-α1/α2 deficiency and persistent pulmonary arterial hypertension of the new-born. The primary characteristics were as follows: (1) increased medial thickness throughout the pulmonary arterial tree and muscle evident in arteries that would normally be non-muscular; (2) hypertrophy rather than proliferation of pulmonary arterial myocytes; (3) no evidence of intimal fibrosis, hyperplasia or plexiform lesions (although plexiform lesions are not common features in mouse models); (4) simplification of alveoli and reduced alveolar numbers, with thickening of alveolar walls and no sign of intra-alveolar or interstitial edema; (5) increases in pulmonary-vascular resistance; (6) right ventricular dilation and increased right ventricular volume/weight ratio; (7) age-dependent increases in right ventricular pressures and decreases in cardiac output; (8) attenuation of hypoxic pulmonary vasoconstriction and thus acute hypoxic pulmonary hypertension; (9) no significant parenchymal fibrosis or lung oedema, indicating that pulmonary vasculopathy arose independent of either left ventricular dysfunction or chronic congestive heart failure[40].

When taken together histological and functional data are most consistent with persistent pulmonary hypertension of the new-born (PPHN) in humans[11,71–74], rather than neonatal pulmonary hypertension secondary to congenital lung abnormalities, such as alveolar capillary dysplasia, congenital diaphragmatic hernia, or bronchopulmonary dysplasia[43]. Furthermore, smooth muscle-selective AMPK-α1/α2 deficiency is associated with hypertrophy of pulmonary arterial smooth muscles, rather than proliferation, compromised alveolarisation, and a lack of intimal fibrosis and plexiform lesions (although plexiform lesions are not a common feature of mouse models)[41–43], thus differentiating the present phenotype from persistent pulmonary hypertension of the adult[38].

It seems likely that pulmonary hypertension and right ventricular decompensation ultimately led to premature death, as there was no evidence of systemic hypertension, vasculopathy nor any other identifiable pathology. Reduced right ventricular fractional shortening was noted and coupled with age-dependent reductions in right ventricular cardiac output, which is most likely explained by increases in pulmonary vascular resistance, because there was no evidence of right ventricular thinning. Supporting this, no similar cardiopulmonary phenotypes have been observed in inducible cardiac-specific AMPK-α1/α2 knockouts without stress[75]. By contrast, observed reductions in left ventricular fractional shortening, left ventricular cardiac output and stroke volume may result from reductions in pulmonary venous return and pre-load, and/or thinning of the left ventricular walls, as this was not associated with systemic arterial hypertension. In this respect it is important to note that similar left, but not right, ventricular dysfunction was observed with smooth muscle-selective AMPK-α2 deletion, consistent with previous observations on cardiac-specific AMPK-α2 knockouts[51] where data suggested a predominant protective role of AMPK-α2 over AMPK-α1 in the heart[51]. In this respect it is also notable that cardiac-specific dual deletion of AMPK-β1/β2 subunits precipitates left-side biased ventricular dilation but with no evidence of the multifocal cardiomyocyte thinning or left ventricular fibrosis[76] observed here. That said, a Glu506Lys substitution in the AMPKγ2 subunit has been linked with a high incidence of right ventricular hypertrophy in humans[77], highlighting the possibility that cell type, AMPK subunit-specific deficiencies and/or context may determine outcomes.

At the cellular level, AMPK-α1/α2 deficiency was also associated with reduced $K_V1.5$ channel availability but not expression in acutely isolated pulmonary arterial myocytes, which has also been indicated in animal models of PPHN[52,53]; representing perhaps another differentiating factor when compared to pulmonary hypertension secondary to congenital heart disease[78]. This was a surprise, given that AMPK-α1 directly phosphorylates and inhibits $K_V1.5$ channel currents in wild-type cells during hypoxia[2,3]. Nevertheless, dual dephosphomimetic mutation of the two $K_V1.5$ serine residues (ser559 and ser592) phosphorylated by AMPK, reduced potassium currents carried by recombinant $K_V1.5$ channels expressed in HEK293 cells, highlighting once more possible cell- and context-specific actions of AMPK that could also impact channel trafficking[79,80] and/or degradation[81–83]. Intriguingly, both S592 and S559 sit within the C terminal region proximal to a variety of other residues that are known to coordinate channel trafficking and surface expression in a manner directed by, for example, palmitoylation[84], oxidative stress[85], KChIP2 interactions[86] and C-terminal PDZ domain interactions[87]. However, further detailed investigations will be required to determine which of these regulators of $K_V1.5$ trafficking and thus cell surface expression is modulated by AMPK-dependent phosphorylation of S559 and S592.

That said, in accordance with a role in the regulation of $K_V1.5$ trafficking by oxidative stress[85], we found that mitochondrial

membrane depolarisation and ROS accumulation was also observed, with no changes in the expression of either COX4i2 or PGC-1α. This is consistent with the known role of AMPK in the governance of mitochondrial biogenesis, integrity and mitophagy[15,18,19], where AMPK has been proposed to serve as a check-point for mitochondrial quality control, initiating mitophagy of fragmented mitochondria to allow efficient and timely apoptosis[88,89]. Accordingly, we observed increased fragmentation of mitochondria in pulmonary arterial myocytes from AMPK-α1/α2 knockouts. This is in-line with previous reports on mitochondrial dysfunction and ROS accumulation being associated with PPHN[23,65,66], and pulmonary hypertension in adults[67–70]. Whether mitochondrial ROS accumulation consequent to mitochondrial dysfunction contributes to causal mechanisms and/or acts as a further deleterious factor promoting persistent mitochondrial and/or nuclear DNA damage remains to be determined[90]. What is clear, however, is that AMPK phosphorylates and regulates many other signaling complexes that have been implicated in pulmonary hypertension[15], and all such actions would be abolished by AMPK-α1/α2 deletion.

In conclusion, the present study provides the first direct evidence that the induction of PPHN after birth may be triggered by AMPK-α1/α2 insufficiency in pulmonary arterial myocytes. Therefore, AMPK-α1 and AMPK-α2 likely impact processes critical to pulmonary vascular development and/or subsequent adaptation to extrauterine life. That similar pathologies were not observed with either AMPK-α1 or AMPK-α2 deletion alone is also important, because this suggests redundancy of function that might afford new AMPK isoform- and thus pulmonary-selective therapeutic strategies against PPHN. Indeed, amelioration by metformin of PPHN[23–25] and pulmonary hypertension in the adult[5,27,91,92] has been noted[26], which may be mediated, at least in part, through AMPK activation as opposed to reduced efficiency of redox and electron transfer at mitochondrial complex I per se[30] or inhibition of fructose-1,6 bisphosphatase[31]. Given that metformin is highly efficacious against type 2 diabetes, it is also interesting to note that AMPK deficiency[93], PPHN[94–96] and pulmonary hypertension of the adult[97,98] are associated with obesity and type 2 diabetes (maternal with PPHN), not least because similar signs and symptoms are present even where no such comorbidity is evident[99–101]. Further studies are therefore warranted and required to identify the critical determining pathways by which AMPK deficiency confers such selective cardiopulmonary dysfunction after birth and thus PPHN.

## Methods
### Breeding and genotyping of transgenic mice
Global, dual deletion of the genes encoding AMPK-α1 (*Prkaa1*) and AMPK-α2 (*Prkaa2*) is embryonic lethal. We therefore employed conditional deletion of AMPK-α1, AMPK-α2 or both AMPK-α1/α2, using mice in which the sequence encoding the catalytic site of either or both of the α subunits was flanked by loxP sequences[32–34]. To direct AMPK deletion to myocytes across the cardiopulmonary system, these were crossed with mice in which Cre recombinase was under the control of the transgelin (smooth muscle protein 22α) promoter (The Jackson Laboratory, Bar Harbor, ME, USA [stock number 017491, Tg (Tagln-cre) 1Her/J in C57BL/6:129SJL background]). Although atrial and ventricular myocytes do not express transgelin in the adult, transient developmental expression of transgelin is observed in these cells[35]. Therefore, with our conditional deletion strategy genomic recombination may occur in both cardiomyocytes and arterial smooth muscles. Transgelin-Cre mice do not, however, exhibit Cre expression in endothelial cells[35], which provides the specificity of deletion required for our studies. We detected the presence of wild-type or floxed alleles and Cre recombinase as described previously[2].

We used two primers for each AMPK catalytic subunit: α1-forward: 5′ TATTGCTGCCATTAGGCTAC 3′, α1-reverse: 5′ GACCTGACAGAAT AGGATATGCCCAACCTC 3′; α2-forward 5′ GCTTAGCACGTTACCCTGG ATGG 3′, α2-reverse 5′ GTTATCAGCCCAACTAATTACAC 3′.

For the detection of Cre recombinase we employed: TH3, 5′-CTTTCCTTCCTTTATTGAGAT-3′, TH5, 5′-CACCCTGACCCAAGCACT-3′ and Cre-UD, 5′-GATACCTGGCCTGGTCTCG-3′.

Wild-type or floxed alleles and Cre-recombinase were detected using a standard PCR protocol for all genotype primers: 92 °C for 5 min, 92 °C for 45 s, 56 °C for 45 s, 72 °C for 60 s, and 72 °C for 7 min for 35 cycles and then 4 °C as the holding temperature. 15 µl samples were run on 2% agarose gels with 10 µl SYBR®Safe DNA Gel Stain (Invitrogen) in TBE buffer against a 100 bp DNA ladder (GeneRuler™, Fermentas) using a Model 200/2.0 Power Supply (Bio-Rad). Gels were imaged using a Genius Bio Imaging System and GeneSnap software (Syngene).

All experiments were performed in accordance with the regulations of the United Kingdom Animals (Scientific Procedures) Act of 1986. All studies and breeding were approved by the University of Edinburgh College of Medicine Animal Welfare and Ethics Review Committee and performed under UK Home Office project license held by AME (PBA4DCF9D). Both male and female mice were used, all of which were on a C57/Bl6 background. Numbers of mice (≥3 per measure) used are indicated for each experiment.

### Isolation of pulmonary arterial smooth muscle cells
Mice were sacrificed by cervical dislocation. The heart and lungs were removed en bloc and single smooth muscle cells were isolated from 2nd and 3rd order branches off the main intrapulmonary artery as described previously[2].

### Histology and immunohistology
Hearts, lungs, livers, kidneys and brains were fixed in a phosphate-buffered solution with 4% paraformaldehyde (Sigma) and kept at 4 °C. All the hearts were trimmed for histology by cutting them longitudinally, ensuring all the cardiac cavities were included (i.e., through the mid-line of both atrial appendages). Tissues were then embedded in paraffin blocks using a Thermo Electron Excelsior tissue processor (Thermo, UK). This was achieved by serial immersion in the following: 70% ethanol (1 h), 90% ethanol (1 h), absolute ethanol (1 h ×4), xylene (1 h ×2), wax (1.3 h × 3). The blocks were then sectioned at 4 µm and placed on a glass slide, which was then incubated at 52 °C for at least an hour. Tissue sections were then dewaxed and rehydrated using an autostainer (ST5010 Autostainer XL, Leica Microsystems, UK) through 3 × 5 min cycles with xylene, 2 × 3 min cycles with 100% ethanol, 1 × 2 min cycles with 95% ethanol and 1 × 5 min wash with distilled $H_2O$ (d$H_2O$). Sections were then left to stand in d$H_2O$ until the next procedure. For the following steps all reagents were from Thermo Fisher Scientific (UK), unless stated otherwise. For haematoxylin and eosin (H&E) staining after rehydration, slides were processed in the same autostainer by sequential immersion in haematoxylin (5 min), d$H_2O$ (5 min), Scott's tap water (2 min), d$H_2O$ (5 min), and eosin (3 min). For Masson's trichrome stain, slides were processed in one batch by serial immersion in: Bouin's solution at 60 °C (1 h), running tap water (20 min), Celestine blue (10 min), and Harris Haematoxylin (5 min). For Picrosirius Red stain, slides were processed in one batch by serial immersion in: Weigert's haematoxylin (10 min) and running tap water (20 min). At this point, slides were incubated with Picrosirius Red until equilibrium (1 h). At the end of each staining, slides were rinsed in 1% acetic acid, followed by absolute ethanol. Stained slides were dehydrated as follows: d$H_2O$ wash (45 s), 70% ethanol (30 s), 95% ethanol (2 x 30 s), 100% ethanol (2 x 1 min), ethanol/xylene (1 min), and xylene (3 x 1 min). Coverslips were applied using Pertex Mounting Medium (CellPath Ltd, UK).

### Immunolabelling for Ki67 and α-smooth muscle actin
Lung sections were stained using an envision+ rabbit kit (Agilent-Dako UK), mouse anti-Ki67 (Clone MiB1, Agilent-Dako UK) and rabbit anti-α-smooth muscle actin antibody (ab5694, Abcam UK). Antigen retrieval

was achieved by pre-treatment with citrate at pH 6 and 110 °C for 5 min. Sections were then incubated with the primary antibody (1:400 dilution) overnight at 4 °C.

## Medial wall area and muscularisation

Double-blind analysis of sections was completed by light microscopy, with small vessels (20–150 μm outer diameter) from the left lobe categorized as muscular, partly muscular or non-muscular as previously described[102]. The outer diameter and the area stained with α-smooth muscle actin were measured using ImageJ (Ver 1.41, NIH, Bethesda, MD, USA). ~30 vessels from 4–7 different fields (20x) were analysed from each individual animal. The following indices were used to quantify vascular remodelling: (1) medial thickness (cross-sectional medial wall area ÷ diameter); (2) percentage medial wall thickness [cross-sectional medial wall area ÷ (total cross-sectional area × 100)].

## Alveolar count and alveolar wall thickness

The number of alveolar spaces per field were counted and presented as the number of airspaces per millimetre square area. Alveolar wall thickness measurements were obtained by direct measurement of 60–70 alveolar walls for each mouse from at least three fields (40x) using NDP view 2 software (Hamamatsu Photonics Ltd, U.K.), as described previously by others and following American Thoracic Society guidelines on quantitative assessment of lung structure[103,104].

## Heart weights

Atria were trimmed, and right ventricular remodelling was measured by weighing the right ventricle (RV) relative to the left ventricle plus septum (LV+S) and the RV or LV+S relative to the animal's body weight (BW).

## Laser microdissection and quantitative RT-PCR

Dissected pulmonary arteries were immersed in OCT embedding compound and quickly frozen in a bath of 100% ethanol and dry ice. Arterial ring sections were cut at a thickness of 5 μm using a cryostat set to −20 °C and mounted onto Polyethylene Naphthalate (PEN)-membrane covered slides (Zeiss). Each slide was air dried for 5 min and then briefly submerged in water, stained with haematoxylin and eosin, and briefly washed by submerging in water and 100% ethanol. Slides were then immediately used for non-contact laser capture microdissection using the PALM MicroBeam system (Zeiss). The energy and focus settings for the laser were adjusted for each individual slide to ensure accuracy in the cutting of the sample. Several regions of interest were drawn per sample that encircled the smooth muscle layer of the artery, and multiple samples were dissected (2–8 mm$^2$) from at least four (4) arterial rings per mouse. Samples were captured into the same adhesive cap of a 0.2 ml AdhesiveCap 200 opaque (Zeiss) collection tube and stored on dry ice or kept at −80 °C until further processing.

For RNA isolation, the Qiagen RNeasy Micro Kit protocol was followed at room temperature as per manufacturer's instructions (https://www.qiagen.com/gb/resources/resourcedetail?id=682963a5-737a-46d2-bc9f-fa137b379ab5&lang=en). Briefly, buffer containing 0.01% β-Mercaptoethanol and 20 ng carrier RNA was added to each sample and vortexed. The whole sample volume was transferred to an RNeasy MinElute spin column (Qiagen) and after a series of buffer washes, RNA was eluted into a DNA- and RNA-free 1.5 ml collection tube using RNase-free water and immediately placed on ice. A Nano-Drop 1000 photospectrometer (Thermo Fisher Scientific) was then used to determine the concentration of RNA for each sample (1–13 ng/ml). Sample eluates were converted into cDNA using the Transcriptor High Fidelity cDNA synthesis Kit (Roche) following the manufacturers' instructions.

For qPCR analysis, 2 μl of cDNA in RNase free water was made up to 20 μl with FastStart Universal SYBR Green Master (ROX, 10 μl, Roche), Ultra Pure Water (6.4 μl, SIGMA) and forward and reverse

primers for AMPK-α1 (Fwd: gtaccaggtcatcagtacacca; Rev: gtggaccac-catatgcctgt; amplicon size: 175 bp), and AMPK-α2 (Fwd: tgatgca-catgctccaggtg; Rev: catcgtaggaggggtcttca; amplicon size: 120 bp). Primers for transgelin were from the QuantiTect Pimer Assay (Qiagen, QT00165179; amplicon size 102 bp). The sample was then centrifuged and 20 μl added to a MicroAmpTM Fast Optical 96-Well Reaction Plate (Greiner bio-one), the reaction plate sealed with an optical adhesive cover (Applied Biosystems) and the plate centrifuged. The reaction was then run on a sequence detection system (Applied Biosystems) using AmpliTaq Fast DNA Polymerase, with a 2 min initial step at 50 °C, followed by a 10 min step at 95 °C, then 40 × 15 s steps at 95 °C. This was followed by a dissociation stage with 15 s at 95 °C, 20 s at 60 °C and 15 s at 95 °C. Positive controls included RNA extracted from the whole artery of an AMPK floxed mouse. Negative controls included cell aspirant but no reverse transcriptase added, and aspirant of extracellular medium. None of the negative controls produced any detectable amplicon, ruling out genomic or other contamination.

## Quantitative RT-PCR on whole arteries without endothelium

For qRT-PCR on endothelium denuded pulmonary arteries, RNA was extracted using the High Pure RNA Tissue Kit (Roche) following the manufacturer's guidelines and the concentration determined using the Nanodrop 1000 spectrophotometer (ThermoScientific). cDNA synthesis was carried out using the Transcriptor High Fidelity cDNA synthesis Kit (Roche) following the manufacturers' instructions. For qPCR analysis, 2.5 μl of cDNA in RNase free water was made up to 25 μl with FastStart Universal SYBR Green Master (ROX, 12.5 μl, Roche), UltraPure Water (8 μl, SIGMA) and fwd and rev primers (QuantiTect) for the genes encoding K$_V$1.5 (*Kcna5*), TASK-1 (Kcnk3), PGC-1α (*Ppargc1a*) and COX4I2 (*Cox4i2*). Samples were then centrifuged and 25 μl added to a MicroAmp$^{TM}$ Fast Optical 96-Well Reaction Plate (Greiner bio-one), the reaction plate sealed with an optical adhesive cover (Applied Biosystems) and the plate centrifuged. The reaction was then run on a sequence detection system (Applied Biosystems) using AmpliTaq Fast DNA Polymerase, with a 2 min initial step at 50 °C, followed by a 10 min step at 95 °C, then a 15 s step at 95 °C which was repeated 40 times. This was followed by a dissociation stage with a 15 s step at 95 °C, a 20 s step at 60 °C and a 15 s step at 95 °C. Data were normalised to a housekeeping gene (*Ipo8*). Negative controls included control cell aspirants for which no reverse transcriptase was added, and aspiration of extracellular medium and PCR controls. None of the controls produced any detectable amplicon, ruling out genomic or other contamination.

## Doppler ultrasound

Mice were rendered anesthetised using Isoflurane, induced at level 5 and set to level 2 for procedures. Isoflurane was mixed with 21% O$_2$ set at 1 L/min for normoxia and 8% O$_2$ for the hypoxic challenge (0.05% CO$_2$, balanced with N$_2$). Using the Vevo770 (Visualsonics Inc) and RMV 707B transducer with a centre frequency of 30 MHz, a Parasternal Long Axis (PLAX) view of the heart of anaesthetised mice was obtained, then modified to bring the pulmonary artery into view for spectral wave Doppler measurements. Peak velocity (mm/s) and velocity time integral (VTI, cm) were estimated from systolic waves to provide a non-invasive approximation of changes in pulmonary vascular resistance as described by others[45,105], and this was confirmed here by induction of pulmonary selective vasoconstriction through airway hypoxia[4]. Measurement of spectral Doppler velocities were taken under normoxia (21% O$_2$; 0.05% CO$_2$ balanced with N$_2$) and during an acute hypoxic challenge (8% O$_2$; 0.05% CO$_2$ balanced with N$_2$) of ≈2 min duration followed by a return to normoxia.

Due to the base-to-apex contraction of the RV it is difficult to accurately quantify RV function other than by fractional shortening, which was therefore assessed using a 2D echocardiographic measurement[49,106]. An M-mode across the anterior and posterior left

ventricle walls was used to determine fractional shortening of the left ventricle from a standard PLAX view. To determine RV fractional shortening an M-mode was analysed using ultrasound taken across the right ventricle free wall and the septum from a modified PLAX view[105].

## Mitochondrial imaging

Mitochondrial volumes and membrane potentials were assessed by fluorescence microscopy using: (1) 100 nmol/L Mitotracker Green (490 nm excitation; 516 nm emission; MitoTracker® Green FM, M7514, Thermo Scientific, UK) and 100 nmol/L TMRE (Tetramethylrhodamine, ethyl ester; 549 nm excitation; 575 nm emission; ab113852, Abcam UK); (2) 300 nmol/L Mitotracker Red (581 nm excitation; 644 nm emission; MitoTracker™ Red FM, M22425, Thermo Scientific, UK) and 10 μg/ml Rhodamine 123 (507 nm excitation; 529 nm emission; R8004, Sigma-Aldrich Company Ltd, UK). Measures of reactive oxygen species were obtained using 1 μmol/L MitoSOX (510 nm excitation; 580 nm emission, MitoSOX™, M36008, Thermo Scientific, UK). In each case cells were incubated in PSS at 37 °C for 30 min. Whole-cell Rhodamine 123 fluorescence was analysed before (baseline) and under specified conditions for a maximum of 15 min. Non-quenching conditions were confirmed at the end of every experiment by adding the mitochondrial uncoupler carbonyl cyanide 4-(trifluoromethoxy) phenylhydrazone (FCCP; 10 μmol/L, C2920, Sigma-Aldrich Company Ltd, UK). Images of emitted fluorescence were acquired at 37 °C and 0.1 Hz, using a Zeiss Fluar 40 × 1.3 n.a. oil immersion objective. The microscope was coupled to a Hamamatsu Orca CCD camera and Sutter DG5 light source with appropriate filter sets. Analysis was completed using Volocity 5.5.1 image acquisition and analysis software (Perkin-Elmer, UK). For the detection of mitochondrial clusters, the intensity threshold was systematically set to 27% of the maximal fluorescence with MitoTracker Green for each individual cell. This acquisition and analysis system is able to accurately resolve elements of labelling smaller than 0.2 μm in size in each of the X-, Y- and Z-planes. However, the precisely controlled environment under which these 'optimal' measurements are obtained cannot be recreated under our experimental conditions. Therefore, we set a more conservative value on the limit of resolution and only included those volumes of labelling measuring ≥0.5 μm in the X-, Y- and Z-planes (volume ≥ 0.125 μm³) with any element of labelling smaller than these limits excluded from consideration[107].

## Fibrosis staining and analysis

For semi-automated analyses of fibrosis, coated slides were overlaid with 4 μm thick sections of paraffin embedded tissue and stained with Picro-Sirius Red (PRS) using standard methods. The slides were then scanned with a Hamamatsu nanozoomer-XR scanner (Hamamatsu, UK), using standard brightfield settings. Scanned images were opened in QuPath[108], where the entire areas corresponding to the tissue section (e.g., left ventricle and septum) were selected and exported to FIJI[109] for analysis. In the process of selecting areas for analysis, large vessels were excluded from the selection (in order to eliminate bias resulting from inclusion of normal perivascular fibrous tissue in the analyses). These pictures were saved as RGB files (.tiff) with a downsize factor of 3 in Qupath. A macro for PRS detection was then developed using FIJI. The macro developed and separated the RGB images into different channels using a colour deconvolution method, and also produced a quality control image[110]. Briefly, the colour vectors (i.e., PRS, counterstain and non-tissue background) were determined manually using regions of interest (ROI) in a picture of a tissue sample (e.g., myocardial, lung) with moderate fibrosis. The RGB values obtained for each colour were then recorded in a macro that was used to analyse the complete dataset by producing the following values in μm²: total tissue surface (i.e., PRS and counterstain) and total PRS stained surface. The PRS stain was then expressed as a percentage of the total tissue surface and assessed by Kruskal–Wallis Test (%PRS stain versus Group; all images were assessed individually for measurement accuracy).

## Electrophysiology

Potassium currents were recorded by whole-cell patch clamp as described previously[2] and assessed using voltage ramps (−60 to +60 mV; HEK293 cells) and single (−60 to +50 mV; HEK293 cells) or multiple (−80 to +0 mV, 10-mV increments; pulmonary arterial myocytes) voltage steps. Recordings were conducted at 37 °C. The standard perfusate (pH 7.4) was composed of 135 mM NaCl, 5 mM KCl, 1 mM MgCl$_2$, 1 mM CaCl$_2$, 10 mM glucose, and 10 mM HEPES (pH 7.4). The pipette solution (pH 7.2) consisted of 140 mM KCl, 10 mM EGTA, 1 mM MgCl$_2$, 10 mM HEPES, 4 mM Na$_2$ATP, and 0.1 mM Na$_3$GTP. Patch pipettes had resistances of 4–6 megohms. Cells were only accepted for analysis if the seal resistance was >3Gohms. Series resistance was monitored and compensated (60 to 80%) after achieving the whole-cell configuration. If a greater than 20% increase occurred during the recording, then the experiment was terminated. Cells were normally superfused at a rate of 1 ml/min when drug treatments were tested. Signals were sampled at 10 kHz and low-pass-filtered at 2 kHz. All experiments on pulmonary arterial myocyte potassium currents were recorded in the presence of paxilline (1 μmol/L), in order to block the large conductance voltage- and calcium-activated potassium channel (KCa1.1). Voltage-clamp acquisition and analysis protocols were performed using an Axopatch 200 A amplifier/Digidata 1200 interface controlled by Clampex 10.0 software (Molecular Devices, Foster City, CA). Offline analysis was performed using Clampfit 10.0 (Molecular Devices, Foster City, CA).

To test the effects of hypoxia, cells were superfused at 3 ml/min with bath solution steadily bubbled with 95% N$_2$/5% CO$_2$ [hypoxia = 6.2 ± 0.3% O$_2$ in the experimental chamber, as measured with an optical oxygen meter (FireStingO2, PyroScience)].

## HEK293 cell transfection

HEK293 cells were cultured in Dulbecco's modified Eagle's medium supplemented with 10% (v/v) fetal bovine serum and 1% (v/v) penicillin/streptomycin. Cells were transfected with 4 μg of pcDNA3.1 encoding a WT or mutant human K$_V$1.5 (KCNA5) using 10 μl of Lipofectamine 2000 (Thermo Fisher Scientific) and used 24–48 h later.

## Right ventricular pressure

Mice were anesthetised with 2.0% isoflurane administered via a non-invasive nose-cone. In the supine position, the right external jugular vein was surgically isolated. Silk ties were placed at the distal ends of the vessel while overhand loops were placed at the proximal ends with 7.0 nylon. A Millar SPR-1000 Mikro-Tip® mouse pressure catheter (1F), an interface cable AEC-10D and independent consoles were used for right ventricular recordings via a PowerLab 4/35 (ADInstruments, Oxford, UK). Prior to insertion, conductance catheter calibration was performed and then zeroed in saline. A transverse venotomy was performed using a micro lancet at the proximal end of the external jugular vein. The catheter was advanced through the superior vena cava and right atrium into the right ventricle (RV), at which point right ventricular pulse waves could be identified. Records of right ventricular pressure (RVP) were taken for at least 30 s once the baseline was stable. In each mouse 15 consecutive RV waves were analysed and averaged to estimate the values for systolic and diastolic RVP.

## Systemic blood pressure measurements

Blood pressure was recorded using a CODA non-invasive blood pressure system (Kent Scientific, USA) as described previously[111].

## Statistical analysis

Where feasible we followed the standards and methodological rigor in pulmonary arterial hypertension as laid out in Circulation Research[112]. GraphPad Prism 6 was used for evaluation of normal distribution and statistical comparisons of mean ± SEM, using parametric or non-parametric analysis as appropriate. Single comparisons were done

with the Mann–Whitney test or by unpaired or paired *t*-tests. Kruskal–Wallis or One-way ANOVA with Dunnet post hoc analysis were used for multiple comparison's tests. P values ≤0.05 were considered significant. All the results presented in this manuscript were repeated and reproduced in at least in three independent experiments.

## Reporting summary

Further information on research design is available in the Nature Research Reporting Summary linked to this article.

## Data availability

All data points are presented in the manuscript and/or supplementary information. Any further information required will be made available by the corresponding author upon reasonable request.

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

## Acknowledgements

First and foremost, Mark Evans would like to acknowledge the late Professor Sheila Glennis Haworth CBE who, during a chance meeting at PVRI 2019, confirmed his conclusion that the pathology described here was most consistent with PPHN, gave further guidance and re-introduced him to Dr Alison Hislop, who all authors thank for her kind consideration and further guidance. We thank Brendan Corcoran for introducing Jorge del Pozo to the Evans Laboratory, without whose kind support and histological expertise we could not have been completed this study. Thanks also to Dr Fiona A. Ross, Professor D. Grahame Hardie and Dr Mark L. Dallas for experimental support. Finally, all authors would like to thank members of the University of Edinburgh Bioresearch and Veterinary Services (BVS) who stepped above and beyond their normal duties to help us achieve our aims by facilitating studies on neonatal mice, namely Matt Sharp, Neil Odey, Peter Rutherford and Iain McCall. This work was funded by programme grants awarded to AME from the Wellcome Trust (WT081195MA, which funded S.L.) and the British Heart Foundation (RG/12/14/29885, which funded J.M.S., S.L. and H.M.). Additional support was provided by Professor Michael J. Shipston, Dean of Biomedical Sciences, University of Edinburgh.

## Author contributions

A.M.E. conceived of this study and wrote the manuscript. J.M.S. and J.D.P. provided detailed and insightful feedback. J.M.S., J.D.P. and A.M.E. developed the figures. J.M.S., S.L. and A.M.E. carried out ultrasound experiments and analysis. J.M.S. and A.M.E. carried out mitochondrial imaging and analysis. J.M.S. and A.M.E. carried out electrophysiology and analysis. J.D.P., J.M.S. and A.M.E. carried out histology and analysis.

S.L., J.M.S., H.M. and A.M.E. completed qPCR. S.M. and J.M.S. carried out laser microdissection and end-point PCR. S.M. carried out blood pressure measurements. M.M. and J.M.S. carried out right ventricular pressure measurements and anlaysis. B.V. and M.F. developed and suppled the Prkaa1 and Prkaa2 floxed mice. S.M., J.M.S. and A.M.E. bred and genotyped conditional knockouts.

## Competing interests

The authors declare no competing interests.
