## [Peer Review File · Nature Communications]

Reviewers' Comments:

Reviewer #1:

Remarks to the Author:

This study investigated whether the pulmonary vascular function is differentially affected by dual deletion of AMPK- α 1 and AMPK- α 2 subunits in smooth muscles when compared to a loss of either AMPK- α 1 or AMPK- α 2 alone. The authors observed that dual AMPK- α 1/ α 2 deletion in smooth muscles resulted in increased muscularization and remodeling throughout the pulmonary arterial tree and reduced alveolar numbers, and premature death. Pulmonary hypertension and attenuated hypoxic pulmonary vasoconstriction were observed in AMPK- α 1/ α 2 knockouts. In acutely isolated pulmonary arterial smooth muscles, the authors observed that AMPK- α 1/ α 2 deletion was associated with marked reductions in voltage-gated KV1.5 potassium current amplitude during normoxia, loss of KV1.5 current inhibition during hypoxia, mitochondrial fragmentation, and accumulation of reactive oxygen species. Finally, they observed age-dependent elevation of right ventricular systolic and diastolic pressures, alongside right ventricular dilation and reduced ventricular fractional shortening. None of these outcomes were recapitulated by AMPK- α 1 or AMPK- α 2 deletion alone. They conclude that AMPK- α 1/ α 2 deficiency in smooth muscles promotes persistent pulmonary hypertension of the newborn. The study addressed an important scientific question. All experiments were carefully performed, results are clearly presented, and some conclusion is supported by their data. However, many concerns exist.

1. The authors presented clear evidence that significant pulmonary circulatory pathology develops only in AMPK- α 1/ α 2 double knockout animals. However, the study offers no explanation why this is the case. Single knockout does not result in any change, excluding additive or synergistic effect. The results demonstrating both isoforms are required to maintain normal pulmonary vascular structure/function argue against the authors' statement that "this suggests redundancy of function that might afford new AMPK isoform- and thus pulmonary-selective therapeutic strategies against PPHN".

2. In AMPK- α 1/ α 2 double knockout animals, the authors demonstrated similar structure/function alteration as that observed in PPHN. However, these data are insufficient to support the authors' conclusion that "the present study provides the first direct evidence that the induction of PPHN after birth may be triggered by AMPK- α 1/ α 2 insufficiency in pulmonary arterial myocytes". A human newborn refers to a baby from birth to 2 months of age. Mice at age of 7 weeks are young adults, absolutely not newborns. In order to link their findings with PPHN, mice at a much younger age (i.e., <1 week) must be studied.

3. The authors observed clear evidence of left ventricular structure/function changes in double knockout animals and attributed these alterations to reductions in pulmonary venous return and pre-load, and/or thinning of the left ventricular walls. One critical possibility must be determined. The authors showed no loss of AMPK- α 1 or AMPK- α 2 protein in the left ventricle by Western blot. However, their knockout strategy will certainly delete both isoforms of AMPK in coronary vascular smooth muscle cells and likely alter coronary circulatory function. Coronary flow and coronary flow reserve must be determined.

4. The authors recognize that a previous study demonstrated transgeline-Cre mice exhibit transient developmental expression in the atrial and ventricle. They determined AMPK- α 1 and AMPK- α 2 protein expression in the left ventricle and showed no protein loss. The atrial expression should also be determined as AMPK malfunction is linked with arrhythmias, which may account for some of the animal death.

5. Does knockout one isoform alter the expression and/or activity of another isoform?

4. Ref 2 and 32 are a duplicate.

Reviewer #2:

Remarks to the Author:

This is an interesting paper from well established investigators.

I have several required experiments and major issues with the present data

1) A) Fig 1 please confirm RV abnormalities by measuring BNP and ANP levels. I would also like to see confirmation of histological measurement of RV fibrosis by coll 1 and 3 measurements,

fibronectin SMA levels and fibroblast proliferation. Rv decompensation characterization is critical in the present paper as authors claim that the dual AMPk deletion cause the death by PHT. Omura et al paper circulation 2020 on RV decompensation should be used as model of investigation of RV failure.

B) Authors need to measure RV capillary density and distal coronary artery remodeling. Both are important feature in RV failure see <https://doi.org/10.1161/CIRCULATIONAHA.115.016382> and <https://doi.org/10.1161/ATVBAHA.117.309156>. Both should be discussed.

2) provide on lung section cleaved caspase3 immunofluorescence data to quantify apoptosis levels.

3) hemodynamic measurements in the mice are incomplete please provide cardiac output data. Acute hypoxic response are interesting. Please provide LKB1 levels in the mice as LKB1 in response to hypoxia is suppose to activate AMPKa 1 and a2 to trigger HPV.

4) effects on Kv1.5 are interesting too, does Kv1.5 expression is affected ? please provide quantification of the expression levels of the Key K⁺ channels in PAH (TASK1 , BKCA....) DOI: 10.1183/13993003.00798-2015

5) Mitochondrial functions : Authors showed decreased TMRM (suggestion depolarization) and increase mitosox. Archer and Michelakis suggest that in PAH mitochondria are hyperpolarized resulting in decrease ROS and repress Kv1.5. How the authors explain these differences ?

6) The decrease in Kv1.5 is associated with increase intracellular calcium leading to the activation of key transcription factor like NFAT or RUNX2 <https://doi.org/10.1073/pnas.0610467104> and <https://doi.org/10.1164/rccm.201512-2380OC> please provide quantification of NFAT and RUNX2 activation

7) Given the importance of AMPK on metabolism does the authors seen any changes in cellular metabolism ?

Reviewer #3:

Remarks to the Author:

There are paradoxes in the field surrounding the effects of AMP-activated protein kinase K (AMPK) on hypoxic pulmonary vasoconstriction (for which AMPK is required) versus pulmonary hypertension (linked to AMPK deficiency). In the present study, the authors deleted both AMPK-alpha1 and 2 in smooth muscle only. The authors conclude that this dual deficiency leads to persistent pulmonary arterial hypertension after birth. There are many different experiments and techniques covered in the manuscript and a large volume of data. However, the end result is inconclusive. Several of the main findings are inconclusive, based on extrapolations, and this reviewer did not feel that the paradox posed at the start was adequately resolved. Much of the data represented a characterization, without giving valuable insight into molecular mechanisms. The confusing Kv1.5 results were inconclusive and the explanation given is not a satisfactory resolution. Given all this, the report is an incremental advance that does not provide a conclusive disease mechanism, and is more appropriate for a subspecialty journal. It adds some new information in this subspecialty without providing convincing conclusions of sufficient importance to appeal to a broader readership.

Specific points:

1) The statistical analyses as presented are not satisfactory. The reporting in the figures is inconsistent, with n values being given for some datasets and not others. Also, often the asterisk keys to different P values did not correspond to the numbers of asterisks within the figure panels. In at least one case a P value was given as a number rather than an asterisk. Exact P values are preferable, but in any case the authors should be consistent. When representative images are shown, the n should be quoted. Finally, statistical analysis was not reported for some of the datasets and this should be addressed.

2) Figure 7B - in the scatter plot, inhibition is given in terms of pA/pF. Typically it would be quoted as a fraction or % of the control, as the raw control current densities vary from cell to cell.

Rebuttal to Reviewer Comments

Reviewer #1 (Remarks to the Author):

This study investigated whether the pulmonary vascular function is differentially affected by dual deletion of AMPK- α 1 and AMPK- α 2 subunits in smooth muscles when compared to a loss of either AMPK- α 1 or AMPK- α 2 alone. The authors observed that dual AMPK- α 1/ α 2 deletion in smooth muscles resulted in increased muscularization and remodeling throughout the pulmonary arterial tree and reduced alveolar numbers, and premature death. Pulmonary hypertension and attenuated hypoxic pulmonary vasoconstriction were observed in AMPK- α 1/ α 2 knockouts. In acutely isolated pulmonary arterial smooth muscles, the authors observed that AMPK- α 1/ α 2 deletion was associated with marked reductions in voltage-gated KV1.5 potassium current amplitude during normoxia, loss of KV1.5 current inhibition during hypoxia, mitochondrial fragmentation, and accumulation of reactive oxygen species. Finally, they observed age-dependent elevation of right ventricular systolic and diastolic pressures, alongside right ventricular dilation and reduced ventricular fractional shortening. None of these outcomes were recapitulated by AMPK- α 1 or AMPK- α 2 deletion alone. They conclude that AMPK- α 1/ α 2 deficiency in smooth muscles promotes persistent pulmonary hypertension of the newborn. The study addressed an important scientific question. All experiments were carefully performed, results are clearly presented, and some conclusion is supported by their data. However, many concerns exist.

1. The authors presented clear evidence that significant pulmonary circulatory pathology develops only in AMPK- α 1/ α 2 double knockout animals. However, the study offers no explanation why this is the case. Single knockout does not result in any change, excluding additive or synergistic effect. The results demonstrating both isoforms are required to maintain normal pulmonary vascular structure/function argue against the authors' statement that "this suggests redundancy of function that might afford new AMPK isoform- and thus pulmonary-selective therapeutic strategies against PPHN".

2. In AMPK- α 1/ α 2 double knockout animals, the authors demonstrated similar structure/function alteration as that observed in PPHN. However, these data are insufficient to support the authors' conclusion that "the present study provides the first direct evidence that the induction of PPHN after birth may be triggered by AMPK- α 1/ α 2 insufficiency in pulmonary arterial myocytes". A human newborn refers to a baby from birth to 2 months of age. Mice at age of 7 weeks are young adults, absolutely not newborns. In order to link their findings with PPHN, mice at a much younger age (i.e., <1 week) must be studied.

3. The authors observed clear evidence of left ventricular structure/function changes in double knockout animals and attributed these alterations to reductions in pulmonary venous return and pre-load, and/or thinning of the left ventricular walls. One critical possibility must be determined. The authors showed no loss of AMPK- α 1 or AMPK- α 2 protein in the left ventricle by Western blot. However, their knockout strategy will certainly delete both isoforms of AMPK in coronary vascular smooth muscle cells and likely alter coronary circulatory function. Coronary flow and coronary flow reserve must be determined.

4. The authors recognize that a previous study demonstrated transgelin-Cre mice exhibit transient developmental expression in the atrial and ventricle. They determined AMPK- α 1 and AMPK- α 2 protein expression in the left ventricle and showed no protein loss. The atrial expression should also be determined as AMPK malfunction is linked with arrhythmias, which may account for some of the animal death.

5. Does knockout one isoform alter the expression and/or activity of another isoform?

4. Ref 2 and 32 are a duplicate.

Our answers to the points raised are as follows:

1. The authors presented clear evidence that significant pulmonary circulatory pathology develops only in AMPK- α 1/ α 2 double knockout animals. However, the study offers no explanation why this is the case. Single knockout does not result in any change, excluding additive or synergistic effect. The results demonstrating both isoforms are required to maintain normal pulmonary vascular structure/function argue against the authors' statement that "this suggests redundancy of function that might afford new AMPK isoform- and thus pulmonary-selective therapeutic strategies against PPHN".

We disagree with this comment. Neither deletion of AMPK-alpha1 or AMPK-alpha2 alone precipitates pulmonary hypertension. These knockout mice develop normally and live a full lifespan; although hypoxic pulmonary vasoconstriction is blocked by AMPK-alpha1 deletion. Redundancy of function is highlighted by the fact that PPHN is only triggered by deletion of both AMPK-alpha1 and AMPK-alpha2 catalytic subunits, i.e., in some critical way one can substitute for the other. This demonstrates redundancy of function between the two isoforms.

We demonstrate that AMPK-alpha1/2 deletion leads to loss of KV1.5 potassium currents that is a recognised driver of PPHN and pulmonary hypertension in the adult. Moreover, and consistent with observations on PPHN, we identify pronounced mitochondrial dysfunction in pulmonary arterial smooth muscles of AMPK-alpha1/2 knockouts that is not observed following deletion of AMPK-alpha1 or AMPK-alpha2 alone.

2. In AMPK- α 1/ α 2 double knockout animals, the authors demonstrated similar structure/function alteration as that observed in PPHN. However, these data are insufficient to support the authors' conclusion that "the present study provides the first direct evidence that the induction of PPHN after birth may be triggered by AMPK- α 1/ α 2 insufficiency in pulmonary arterial myocytes". A human newborn refers to a baby from birth to 2 months of age. Mice at age of 7 weeks are young adults, absolutely not newborns. In order to link their findings with PPHN, mice at a much younger age (i.e., <1 week) must be studied.

We thank this Reviewer for highlighting this important matter. Histology has now been completed at P10, an age that reflects the neonatal state. Histology identified alveolar simplification in control neonates as one would expect, where there was a significantly lower number of alveoli and significantly greater alveolar wall thickness when compared to adults. Neither measure for P10 control neonates was significantly different from that for P10 AMPK-alpha1/2 knockouts neonates. We did, however, measure increased medial thickness in P10 AMPK-alpha1/2 knockout neonates relative to P10 controls, consistent with the view that loss of smooth muscle AMPK is the driver of hypertrophy of pulmonary arterial smooth muscles and PPHN.

See page 6, paragraph 3 which now reads:

We therefore assessed lung sections from neonates (P10) prior to alveolarisation⁴¹⁻⁴³. As one would expect, when compared to alveoli in lung sections from adults (Figure 3**) the alveoli of P10 AMPK- α 1/ α 2 floxed mice were fewer in number and had thicker walls ($P < 0.05$; Mann-Whitney independent comparisons test). Importantly, there was no significant difference in either alveolar number or alveolar wall thickness between neonatal (P10) AMPK- α 1/ α 2 floxed mice and AMPK- α 1/ α 2 knockouts (**Figure 4A-D**). However, increases in the medial thickness of pulmonary arteries were evident in AMPK- α 1/ α 2 knockouts when compared to AMPK- α 1/ α 2 floxed mice (**Figure 4E**).**

3. The authors observed clear evidence of left ventricular structure/function changes in double knockout animals and attributed these alterations to reductions in pulmonary venous return and pre-load, and/or thinning of the left ventricular walls. One critical possibility must be determined. The authors showed no loss of AMPK- α 1 or AMPK- α 2 protein in the left ventricle by Western blot. However, their knockout strategy will certainly delete both isoforms of AMPK in coronary vascular smooth muscle cells and likely alter coronary circulatory function. Coronary flow and coronary flow reserve must be determined.

This is a significant point, and we thank this Reviewer for raising this matter. We have now completed histology on the coronary vasculature. We find no significant difference in medial

thickness or vessel number following AMPK deletion. These data are shown in Supplementary Figure 13.

See also page 7 last paragraph through page 8 first paragraph, which now reads:

When taken together these data provide indirect support for the view that right ventricular myopathy of AMPK- α 1/ α 2 knockouts is likely driven, at least in part, by the onset of pulmonary hypertension after birth (see Discussion for further details), a view that gains further indirect support from the fact that we found no evidence of differences in either blood vessel number (n=5; Note, non-parametric Mann-Whitney p=0.056) or medial thickness (n = 5) for the right ventricle of AMPK- α 1/ α 2 knockouts when compared to controls (Supplementary Figure 13).

4. The authors recognize that a previous study demonstrated transgeline-Cre mice exhibit transient developmental expression in the atrial and ventricle. They determined AMPK- α 1 and AMPK- α 2 protein expression in the left ventricle and showed no protein loss. The atrial expression should also be determined as AMPK malfunction is linked with arrhythmias, which may account for some of the animal death.

This is important for completeness. We now present Western blots for the atria, see new panel B in Supplementary Figure 2. This shows AMPK- α 1 and AMPK- α 2 protein expression, consistent with the fact that we find no evidence of atrial pathology or dysfunction.

See also page 4, paragraph 1, lines 7-9, which now reads:

Transgeline does, however, exhibit transient developmental expression in atrial and ventricular myocytes³⁵, although no loss of AMPK- α 1 or AMPK- α 2 protein was revealed for the atria or ventricles by Western blot (Supplementary Figure 2).

5. Does knockout one isotype alter the expression and/or activity of another isotype?

There is no evidence that the authors know of to support this proposal, which has little bearing on the present investigation. Here we show that loss of both AMPK- α 1 and AMPK- α 2 is required to induce PPHN.

4. Ref 2 and 32 are a duplicate.

We thank this reviewer for highlighting this error, which has been corrected.

Reviewer #2 (Remarks to the Author):

This is an interesting paper from well established investigators.

I have several required experiments and major issues with the present data

1) A) Fig 1 please confirm RV abnormalities by measuring BNP and ANP levels. I would also like to see confirmation of histological measurement of RV fibrosis by coll 1 and 3 measurements, fibronectin SMA levels and fibroblast proliferation. Rv decompensation characterization is critical in the present paper as authors claim that the dual AMPK deletion cause the death by PHT. Omura et al paper circulation 2020 on RV decompensation should be used as model of investigation of RV failure. B) Authors need to measure RV capillary density and distal coronary artery remodeling. Both are important feature in RV failure see <https://doi.org/10.1161/CIRCULATIONAHA.115.016382> and <https://doi.org/10.1161/ATVBAHA.117.309156>. Both should be discussed.

2) provide on lung section cleaved caspase3 immunofluorescence data to quantify apoptosis levels.

3) hemodynamic measurements in the mice are incomplete please provide cardiac output data. Acute hypoxic response are interesting. Please provide LKB1 levels in the mice as LKB1 in response to hypoxia is suppose to activate AMPKa 1 and α 2 to trigger HPV.

4) effects on Kv1.5 are interesting too, does Kv1.5 expression is affected ? please provide quantification

of the expression levels of the Key K⁺ channels in PAH (TASK1 , BKCA....) DOI: 10.1183/13993003.00798-2015

5) Mitochondrial functions : Authors showed decreased TMRM (suggestion depolarization) and increase mitoxox. Archer and Michelakis suggest that in PAH mitochondria are hyperpolarized resulting in decrease ROS and repress Kv1.5. How the authors explain these differences ?

6) Th decrease in Kv1.5 is associated with increase intracellular calcium leading to the activation of key transcription factor like NFAT or RUNX2 <https://doi.org/10.1073/pnas.0610467104> and <https://doi.org/10.1164/rccm.201512-2380OC> please provide quantification of NFAT and RUNX2 activation

7) Given the importance of AMPK on metabolism does the authors seen any changes in cellular metabolism ?

Our answer to the points raised are as follows:

1) A) *Fig 1 please confirm RV abnormalities by measuring BNP and ANP levels. I would also like to see confirmation of histological measurement of RV fibrosis by coll 1 and 3 measurments, fibronectin SMA levels and fibroblast proliferation. Rv decompensation charactatrization is critical in the present paper as authors claim that the dual AMPk deletion cause the death by PHT. Omura et al paper circulation 2020 on RV decompensation should*

be used as model of investigation of RV failure. B) Authors need to measure RV capillary density and distal coronary arety remodeling. Both are important feature in RV failure see <https://doi.org/10.1161/CIRCULATIONAHA.115.016382> and <https://doi.org/10.1161/ATVBAHA.117.309156>. Both should be discussed.

We have now completed histology on the coronary vasculature. We find no significant difference in medial thickness or vessel number with AMPK deletion.

See also page 7 last paragraph through page 8 first paragraph, which now reads:

When taken together these data provide indirect support for the view that right ventricular myopathy of AMPK- α 1/ α 2 knockouts is likely driven, at least in part, by the onset of pulmonary hypertension after birth (see Discussion for further details), a view that gains further indirect support from the fact that we found no evidence of differences in either blood vessel number (n=5; Note, non-parametric Mann-Whitney p=0.056) or medial thickness (n = 5) for the right ventricle of AMPK- α 1/ α 2 knockouts when compared to controls (**Supplementary Figure 13**).

Please note that we believe that the right ventricular myopathy is sufficiently characterized by the histological data presented. Importantly, several of the additional tests requested would not provide additional information to dissect if the cardiac lesions we noted are secondary to pulmonary hypertension or not, which would be the main question they should address in this model. It is the functional data we provide, together with histological characterisation of the cardiac lesions, and not the histology data only that allows us to state that this lesion is secondary to pulmonary hypertension at least partly (please see text above).

For this reason, we see no need to carry out some of the analyses proposed. We will now address the rationale for this in detail:

Collagen 1 and 3 measurements - These measurements have been addressed by our Picro Sirius red (PRS) quantification. Even though PRS is not specific to a collagen type and will stain collagen 1 and 3, collagen 1 and 3 are the main components of the extracellular matrix and are expected in areas of fibrosis such as those noted in the myocardium of

AMPK DKO mice. The reference kindly proposed by the reviewer used Masson's trichrome for this purpose, which is also non specific for the staining collagen 1 and 3.

Fibronectin - We presume this Reviewer is considering the possibility that fibrosis of the myocardium in the AMPK knockout mice is ongoing (i.e. that there is active deposition of collagen fibrils by fibroblasts at the moment of sampling). If this was the case, one would expect multifocal positive staining around and within the periphery of areas of fibrosis, co-located to areas of fibroplasia at the periphery of mature fibrous tissue islands. The latter were absent in our samples. Also, we do not see how this information will add any meaningful mechanistic data to the characterization of the cardiac lesion, which is chronic in any case.

Fibroblast proliferation – The rationale here is similar to fibronectin, and we presume this question also relates to ongoing fibrosis at the time of sampling. Histologically, fibroblast proliferation is readily recognisable, as it is associated with accumulation of haphazardly arranged, plump spindle cells indicating fibroplasia (i.e. active deposition of fibrous tissue). Such patterns were absent in our samples. For this reason, any changes over and above those reported here would be subtle and add little to this study (these are chronic lesions).

The reference suggested by the reviewer (Omura et al) uses Masson's trichrome to quantify fibrosis (we have used Picro Sirius Red, which is also an accepted method for this measurement), endothelial density by CD31 immunostaining, and morphometric assessment of the cross sectional area of cardiomyocytes. The latter two are methods that are useful to detect subtle myocardial changes secondary to myocardial adaptation during congestive cardiac failure (i.e., during congestive cardiac failure secondary to any cause, there will be cardiomyocyte hypertrophy and therefore reduction in endothelial density as more myocardial cut surface area is now taken up by the hypertrophic cardiomyocytes). We therefore agree on the usefulness of these techniques, but we don't believe they will add useful information to the characterisation of the lesions we have noted in AMPK DKO mice. Myocardial lesions in DKO mice are not subtle, and feature marked ventricular dilation and fibrosis. The latter would result in reduction of endothelial density in any case. In the case of the former, ventricular dilation results in stretching and therefore thinning of cardiomyocytes, which interferes with the usefulness of any measurement of cardiomyocyte cut surface area (i.e., as these cardiomyocytes are longer and thinner as a result of stretching, their cut surface area is not directly comparable to that of control (Flox) tissue for the areas affected – only measurement of their volume would provide an accurate evaluation, and this is not possible with histology).

2) provide on lung section cleaved caspase3 immunofluorescence data to quantify apoptosis levels.

We find no evidence of apoptosis or proliferation, which has been indicated in pulmonary hypertension of the adult. Rather than this, our data suggest smooth muscle hypertrophy, which is a maker of persistent pulmonary hypertension of the newborn (PPHN).

See page 5 paragraph 2, lines 17-23:

It was therefore surprising that we identified little evidence of mitotic activity in the medial layer of pulmonary arteries from AMPK- α 1/ α 2 knockouts, levels of Ki67 labelling being comparable to controls (**Supplementary Figure 7**), and there was no evidence of cellular shrinkage or nuclear fragmentation that would be expected if there was extensive apoptosis^{38,39}. This suggests that increased muscularization was consequent to smooth muscle cell hypertrophy rather than proliferation.

3) hemodynamic measurements in the mice are incomplete please provide cardiac output data. Acute hypoxic response are interesting. Please provide LKB1 levels in the mice as LKB1 in response to hypoxia is suppose to activate AMPKa 1 and a2 to trigger HPV.

We now provide data on cardiac output for the left ventricle (Supplementary Table I) and right ventricle (new Supplementary Figure 12).

See also page 7, paragraph 2, lines 8-10, which now reads:

Accordingly, we observed a significant age-dependent decrease in cardiac output in AMPK- α 1/ α 2 knockouts when compared to AMPK- α 1/ α 2 floxed mice (Supplementary Figure 12).

Measurements of LKB1 have no bearing on this study. LKB1 is constitutively active and supports acute HPV by phosphorylating AMPK- α 1 after AMP (and ADP) induce the necessary conformational change to facilitate Thr172 phosphorylation. LKB1 expression is not determined by AMPK. Therefore, one would not expect any change in LKB1 levels, and if there was this would have no influence on AMPK which is lacking in the double knockouts studied here, or for that matter HPV, which requires AMPK.

4) effects on Kv1.5 are interesting too, does Kv1.5 expression is affected ? please provide quantification of the expression levels of the Key K⁺ channels in PAH (TASK1 , BKCA....) DOI: 10.1183/13993003.00798-2015

We show that Kv1.5 expression is not altered. Furthermore, we demonstrate that TASK1 channel expression remains unaltered and that TASK1 channels are not subject to regulation by AMPK.

We have now elaborated on this matter for the benefit of the reader.

See Results page 8 paragraph 1 and 2, which now reads:

AMPK- α 1/ α 2 deletion reduces Kv1.5 availability in pulmonary arterial myocytes

We next investigated the impact of AMPK- α 1/ α 2 deletion on voltage-gated potassium (K_V) currents in pulmonary arterial smooth muscle cells, because reduced activity and/or expression of K_V1.5 has been identified as a hallmark of persistent pulmonary hypertension in neonates^{52,53} and pulmonary hypertension in adults^{54,55}, where it has been proposed to decrease K⁺ efflux and thus oppose apoptosis⁵⁵⁻⁵⁸. Under normoxia K_V current density was markedly reduced in acutely isolated pulmonary arterial myocytes from AMPK- α 1/ α 2 knockouts (**Figure 8A-B**), measuring 5.6 ± 0.6 pA/pF compared to 11.6 ± 0.9 pA/pF for controls at 0mV (AMPK- α 1/ α 2 floxed, n=11-12, P<0.01). Furthermore, we observed no reduction in K_V current amplitude during hypoxia in myocytes from AMPK- α 1/ α 2 knockouts (**Supplementary Figure 14**), which was in-line with expectations given our previous finding that AMPK- α 1 deletion blocked hypoxia-evoked reductions in K_V1.5 currents and HPV². That reductions in normoxic K_V current magnitude by AMPK- α 1/ α 2 deletion were due to loss of K_V1.5 in particular was demonstrated by reduced sensitivity of available K_V currents to inhibition by the specific K_V1.5 channel blocker DPO-1 (1 μ mol/L; **Figure 8A-C**). By contrast, K_V1.5 current magnitude in pulmonary arterial myocytes from AMPK- α 1 and AMPK- α 2 knockouts was found to be equivalent to controls during normoxia². The discovery of reduced normoxic K_V1.5 currents in myocytes following AMPK- α 1/ α 2 deletion was in itself an unexpected outcome, given that AMPK- α 1 directly phosphorylates and inhibits K_V1.5 in these cells². It was more surprising still, however, in light of the fact that we found no reduction in transcription of the gene encoding K_V1.5 (*KCNA5*) by qRT-PCR, expression measuring 0.94 ± 0.05 for AMPK- α 1/ α 2 knockouts (n = 4) relative to control (AMPK- α 1/ α 2 floxed; n=4). Hence, we investigated the possibility that K_V1.5 availability might be determined by reduced phosphorylation by AMPK of ser559 and ser592 on the K_V1.5 alpha subunit, by incorporating dephospho-mimetic mutations. Transient transfection of HEK293 cells with the mutant K_V1.5^{ser559A/ser592A} conferred K_V currents that were significantly reduced relative to K_V1.5 current amplitude recorded in paired HEK293 cell cultures transfected with wild type K_V1.5 (**Figure 8D**). This presents us with a paradox given that we have previously established that AMPK directly phosphorylates ser559 and ser592 on the K_V1.5 α subunit and thus inhibits voltage-dependent activation of K_V1.5 currents during hypoxia². AMPK may therefore regulate K_V1.5 availability by multiple, context-specific mechanisms. Indirect support for this is provided by our previous studies which showed that the dephosphomimetic mutation s559a alone confers optimal blockade of K_V1.5 current inhibition by AMPK, while the dephosphomimetic mutation s592a delivers more marked reduction of K_V1.5 phosphorylation by AMPK than ser559a² (see Discussion for further details).

The expression of the gene encoding TASK1 potassium channels (*KCNK3*), which has been associated with pulmonary hypertension⁵⁹⁻⁶² and right ventricular dysfunction⁶³ in adults, also remained unaffected following AMPK- α 1/ α 2 deletion, measuring 0.95 ± 0.13 for AMPK- α 1/ α 2 knockouts relative to controls (AMPK- α 1/ α 2 floxed; n=4). We also found that AMPK activation had no effect on potassium currents carried by TASK1 (**Supplementary Figure 15 and Supplementary methods**) and no AMPK recognition sites were identified for this channel (either by SCANSITE4, or an algorithm available on GitHub (https://github.com/BrunetLabAMPK/AMPK_motif_analyzer⁶⁴)). It is therefore unlikely that TASK1 is subject to modulation by AMPK.

See Discussion page 11 paragraph 1, lines 7-13 which now read:

Intriguingly, both s592 and s559 sit within the C terminal region proximal to a variety of other residues that are known to coordinate channel trafficking and surface expression in a manner directed by, for example, palmitoylation⁸⁴, oxidative stress⁸⁵, KChIP2 interactions⁸⁶ and C-terminal PDZ domain interactions⁸⁷. However, further detailed investigations will be required to determine which of these regulators of Kv1.5 trafficking and thus cell surface expression is modulated by AMPK-dependent phosphorylation of s559 and s592.

5) *Mitochondrial functions* : Authors showed decreased TMRM (suggestion depolarization) and increase mitoxox. Archer and Michelakis suggest that in PAH mitochondria are hyperpolarized resulting in decrease ROS and repress Kv1.5. How the authors explain these differences ?

The studies of Archer and colleagues focussed on hypoxic pulmonary vasoconstriction and thus acute hypoxic pulmonary hypertension of the adult. Therefore, the findings in their studies do not relate to persistent pulmonary hypertension of the newborn that is described in the present investigation.

6) *The decrease in Kv1.5 is associated with increase intracellular calcium leading to the activation of key transcription factor like NFAT or RUNX2* <https://doi.org/10.1073/pnas.0610467104> and <https://doi.org/10.1164/rccm.201512-2380OC> please provide quantification of NFAT and RUNX2 activation.

There is no decrease in Kv1.5 expression evident in these mice. Therefore, we see no reason to assess NFAT or RUNX2 activity, which is perhaps most relevant to models of pulmonary hypertension in adult rats.

See also answer to point 4 above.

7) *Given the importance of AMPK on metabolism does the authors seen any changes in cellular metabolism ?*

Changes in mitochondrial membrane potential and ROS suggest marked changes in smooth muscle cell metabolism. However, further detailed analysis of this will require such extensive investigation that this will warrant a separate paper.

See page 9, paragraph 3, lines 18-26, which now reads:

This is in accordance with the finding that mitochondrial dysfunction and/or ROS accumulation are associated with persistent pulmonary hypertension in neonates^{23,65,66}, and pulmonary hypertension in adults⁶⁷⁻⁷⁰. Mitochondrial dysfunction in pulmonary arterial myocytes of AMPK- α 1/ α 2 knockouts was not consequent to changes in the expression of either COX4i2 (see above) or peroxisome proliferator-activated receptor gamma coactivator 1-*alpha* (PGC-1 α ; 0.74 ± 0.13 , n = 4) in AMPK- α 1/ α 2 knockouts relative to AMPK- α 1/ α 2 floxed, but is entirely consistent with the fact that AMPK directly phosphorylates and thus regulates a plethora of targets critical to the governance of mitochondrial biogenesis, integrity and mitophagy^{15,18,19}.

Reviewer #3 (Remarks to the Author):

There are paradoxes in the field surrounding the effects of AMP-activated protein kinase K (AMPK) on hypoxic pulmonary vasoconstriction (for which AMPK is required) versus pulmonary hypertension (linked to AMPK deficiency). In the present study, the authors deleted both AMPK- α 1 and 2 in smooth muscle only. The authors conclude that this dual deficiency leads to persistent pulmonary arterial hypertension after birth. There are many different experiments and techniques covered in the manuscript and a large volume of data. However, the end result is inconclusive. Several of the main findings are inconclusive, based on extrapolations, and this reviewer did not feel that the paradox posed at the start was adequately resolved. Much of the data represented a characterization, without giving valuable insight into molecular mechanisms. The confusing Kv1.5 results were inconclusive and the explanation given is not a satisfactory resolution. Given all this, the report is an incremental advance that does not provide a conclusive disease mechanism, and is more appropriate for a subspecialty journal. It adds some new information in this subspecialty without providing convincing conclusions of sufficient importance to appeal to a broader readership.

Specific points:

- 1) The statistical analyses as presented are not satisfactory. The reporting in the figures is inconsistent, with n values being given for some datasets and not others. Also, often the asterisk keys to different P values did not correspond to the numbers of asterisks within the figure panels. In at least one case a P value was given as a number rather than an asterisk. Exact P values are preferable, but in any case the authors should be consistent. When representative images are shown, the n should be quoted. Finally, statistical analysis was not reported for some of the datasets and this should be addressed.
- 2) Figure 7B - in the scatter plot, inhibition is given in terms of pA/pF. Typically it would be quoted as a fraction or % of the control, as the raw control current densities vary from cell to cell.

Our answers to the points raised are as follows:

1) The statistical analyses as presented are not satisfactory. The reporting in the figures is inconsistent, with n values being given for some datasets and not others. Also, often the asterisk keys to different P values did not correspond to the numbers of asterisks within the figure panels. In at least one case a P value was given as a number rather than an asterisk. Exact P values are preferable, but in any case the authors should be consistent. When representative images are shown, the n should be quoted. Finally, statistical analysis was not reported for some of the datasets and this should be addressed.

We have cross-checked all figures. All "n values" are now given throughout. The asterisks and P values now match, and presentation of these values is now consistent throughout.

The statistical analyses are in line with those recommended by Circulation, see:

109. Provencher S, *et al.* Standards and Methodological Rigor in Pulmonary Arterial Hypertension Preclinical and Translational Research. *Circ Res* **122**, 1021-1032 (2018).

We have also presented alternative approaches to statistical analyses in the supplementary information.

In short, our approaches to statistical analysis are satisfactory.

2) Figure 7B - in the scatter plot, inhibition is given in terms of pA/pF. Typically it would be quoted as a fraction or % of the control, as the raw control current densities vary from cell to cell.
Studies on potassium currents are not typically reported as % of control when one is comparing paired studies on separate cell populations. In the present circumstance normalisation to cell capacitance (pF; index of cell surface area) is the most appropriate method, as this controls for cell surface area.

However, this comment highlighted a deficiency in our data presentation that we have now corrected through the presentation of the full current-voltage relationship for the DPO-sensitive current in pulmonary arterial myocytes.

See Figure 8, new panel C:

C, Current-voltage relationship for DPO-1-sensitive current (current before DPO-1 minus current after DPO-1).

Reviewers' Comments:

Reviewer #1:

Remarks to the Author:

Multiple new experiments have been added. Previous concerns have been addressed.

Reviewer #2:

Remarks to the Author:

The authors have responded nicely to all my comments. Thank you

Reviewer #3:

Remarks to the Author:

The authors addressed to some extent my two specific points but the main concerns with the manuscript that I had outlined were not addressed in their response.

REBBUTAL TO REVIEWERS' COMMENTS

Authors' responses in bold

Reviewer #1 (Remarks to the Author):

Multiple new experiments have been added. Previous concerns have been addressed.

We thank Reviewer 1 for their helpful suggestions and for confirming that we have addressed their previous concerns.

Reviewer #2 (Remarks to the Author):

The authors have responded nicely to all my comments. Thank you

We thank Reviewer 2 for their helpful suggestions and for confirming that we have dealt with their requests in entirety.

Reviewer #3 (Remarks to the Author):

The authors addressed to some extent my two specific points but the main concerns with the manuscript that I had outlined were not addressed in their response.

From our perspective the matters raised have been dealt with in sufficient detail